

1                    **An agent-based model for flood risk warning.**

**Thomas O'Shea[1], Paul Bates[1] and Jeffrey Neal[1]**
**[1]** School of Geographical Sciences, University of Bristol, UK.
**Correspondence**: Thomas O'Shea (t.oshea@bristol.ac.uk)
**Abstract**
This paper presents a new flood risk behaviour model developed using a coupled
Hydrodynamic Agent-Based Model (HABM). This model uses the LISFLOOD-FP Hydrodynamic
Model and the NetLogo (NL) agent-based framework and is applied to the 2005 flood event
in Carlisle, UK. The hydrodynamic model provides a realistic simulation of detailed flood
dynamics through the event whilst the agent-based model component enables simulation
and analysis of the complex, in-event social response. NetLogo enables alternative
probabilistic daily routine and agent choice scenarios for the individuals of Carlisle to be
simulated in a coupled fashion with the flood inundation. Experiments are also conducted
using a novel, 'enhanced social modelling component', comprised of the Bass Diffusion
Model, to investigate the effect of direct or indirect warnings in flood incident response.
From the analysis of these coupled simulations, management stress points, predictable or
otherwise, can be presented to those responsible for hazard management and post-event
recovery. The results within this paper suggest that these stress points can be present, or
amplified, by a lack of preparedness or a lack of phased evacuation measures. Furthermore,
the methods here outlined have the potential for application elsewhere to reduce the
complexity and improve the effectiveness of flood incident management. The paper
demonstrates the influence that emergent properties have on systematic vulnerability and
risk from natural hazards in coupled socio-environmental systems.



## 1. Introduction

Flood hazard, or flood incident, management is a challenge that incorporates aspects of the natural sciences (hydrology, ecology, etc.), the social sciences (economics, politics, psychology, culture, etc.) and engineering. It is important for the efficiency and efficacy of decision-making processes to recognise that decision-making during floods involves what has been termed "technical complexity" (Nunes Correia, Fordham, Da Graca Raravia & Bernardo 1998). Specifically, this is the social response to the hazard, and encompasses interactions between individuals, their decision-making and collective, during-event, behaviours. This complexity cannot, either theoretically or physically, be eliminated when planning for flooding incidents (Assaf & Hartford, 2002; Bennet & Tang, 2017; Correia, Rego, Saravia & Ramos, 1998 and Dawson, Peppe & Wang, 2011) and can be a threat to effective planning processes (Axelrod, 1970; Nunes Correia et al., 1998).  In a broader sense, this complexity is a measure of the scale of the interactions within the affected area, encompassing dynamic multi-scale interactions and adaptions between individuals, groups, infrastructures, government and the economy, all contributing to the social, political and physical aspects of flood hazard management (Dugdale, Saoud, Pavard, & Pallamin, 2009; Fordham, 1992; IPCC, 2014; Kossiakoff & Sweet, 2002; Werrity, Houston, Ball, Tavendale & Black 2007 and Wisner, Blaikie, Cannon & Davies, 1994).

Recent decades have seen strong emphasis being placed on multi-scale, *participatory* methods for dealing with floods resulting in a paradigm shift from *flood defence* to *flood risk management* (Assaf & Hartford, 2002, Dawson et al., 2011, DEFRA, 2007; IPCC, 2014 and Wisner et al., 1994). Such participation means the inclusive involvement of individuals and multiple agencies in the processes of hazard management, policy implementation and post-event recovery. This emphasis is logical in that it aims to incorporate, as far as possible, the requirements of all those involved in the hazard planning process across a scale hierarchy that passes from government bodies to emergency services, and on to the affected individuals themselves. The complexity of such an ideal becomes apparent given that the intricate natures of human environments and environmental dynamics are, to a large degree, perceived as independent, and that when the two come into contact, complexity becomes amplified within a coupled socio-environmental system. For example, between 2010 and 2015, UK Government policy for flooding underwent a transformation that sought to address some of the known complexities of flood incident management (DEFRA, 2007; Eberlen, Scholz & Gagliolo 2017; The Environment Agency, 2012 & 2016). The UK Government's Department for Environment, Food & Rural Affairs (DEFRA) national framework for flood management emphasises the importance of localised decisions about flood risk and makes suggestions for developing community-based solutions to manage flood risk on a finer spatial scale. This transformation emphasised the need for innovative new approaches to managing the localised risk of flooding. This was expected to provide the foundation for better management at the larger scale as 'good practice' innovations spread across more communities. Thus, UK flood policy can be defined as moving from a top-down to bottom-up approach, often referred to as '*alternative action'* (DEFRA, 2007; Kossiakoff & Sweet, 2002).


Whilst both top-down and 'alternative action' bottom-up approaches will be likely to have
divergent outcomes owing to the different emphasis each places on variables within their
respective approaches, the shift towards a bottom-up strategy indicates an
acknowledgement of the need for greater local participation in decision making; something
which is difficult to achieve with the 'black-box' forms of assistance seen in most top-down
approaches (Sabatier, 1986). Conversely, to formulate an effective bottom-up approach, the
dynamics of the individual base elements, which in this model are individual people and are
termed 'agents', must be specified to a relatively intricate degree of detail. This is because
theory suggests individual and grouped responses will have a significant influence on the
dynamics which emerge at higher systematic levels and so accounting for as much detail as
possible at the individual level will have a bearing on the detail that can be developed within
the descriptions of the whole system (Bresser-Pereira, Maravall & Przeworski, 1993 & Müller
et al., 2013). Here, it is believed that Individual and grouped responses are defined by
environmental, inter-personal interaction and interpretation (Alexander, 1980; Assaf &
Hartford, 2002 and Axelrod, 1970) and that these are characteristic behaviours of sub-
systematic processes which are either not present or not considered in, coarser, top-down
models of physical process; despite potentially having a significant influence on the outcome
of an event in which they are involved (Nunes Correia et al., 1998)**.**

Agent Based Models (ABMs), defined as "a computational method for simulating the actions
and interactions of autonomous decision-making entities in a network or system, with the
aim of assessing their effects on the whole system" (Dawson et al., 2011), provide a potential
means to characterise these interactions.  Essentially, this is a form of computerised model
capable of simulating the emergent behaviour of complex systems. In such models,
individuals and organisations are represented as 'agents' within a simulated environment
(Railsback & Grimm, 2012). In recent years there has been a proliferation of ABM applications
within the research community and examples of these applications relevant to flooding
encompass: (i) the role of social media in flood evacuation processes (Du, Cai & Sun 2017); (ii)
human perception, understanding and anticipation of flash floods (Morss, Mulder, Lazo &
Demuth, 2016; Narsizi, Mysore, & Mishra, 2006,); and (iii) the effectiveness of simultaneous
and staged flood evacuation strategies (Chu, 2015; Dawson et al., 2011; Zarboutis &
Marmaras, 2005).  A key issue for such applications is the development of realistic flooding
scenarios to drive the behaviour of the modelled agents.

Hydrodynamic models can produce this information so long as they are developed with high
quality terrain and boundary condition information (see for example (Neal, Schumann &
Bates, 2012)), but to date ABM applications have not taken advantage of the latest
developments in flood inundation modelling.  The only study to date to drive an ABM with a
hydrodynamic model was that of Dawson (et al., 2011). Here a simple diffusive wave model
which solves Manning's equation over a raster grid of cells was implemented within an ABM
to simulate a coastal flood and showed considerable potential. However, this study initially
coded the hydrodynamic model directly within the ABM meaning advantage could not be
taken of recent developments in efficient numerical methods for solving the shallow water
equations (Bates, Horrit & Fewtrell, 2010) and high-performance computing (*e.g.* Neal,


Fewtrell, Bates & Wright 2010) architectures. As a result, computational costs were high, and this limited the domain size and resolution of the modelling that could be undertaken. Instead of directly embedding the hydrodynamic model within the ABM, a more pragmatic solution is to indirectly couple a separate, and highly optimized, hydrodynamic model with an existing ABM framework. This would allow each code to be properly optimized for the task it performs and enable each to be more easily updated as new methods become available. This is the objective of this paper, where we develop such a coupled hydrodynamic model/Agent-Based model framework (hereafter termed a Hydrodynamic Agent-Based Model, or HABM) and use this to address two currently unresolved questions relating to flood evacuation warnings. These two specific questions are:

1. During a flood, does the site-specific urban topography and morphology change the optimum evacuation warning strategy?

2. Do people (agents) respond better to direct or indirect (word of mouth) evacuation warnings for a flood event?

To date research on flood warnings and evacuation has examined the challenges and changes in thinking required to tackle the paradox of flood 'control' (Wisner et al., ch 6, 2015), the dynamic approaches required to address different forms of flood event (Berendracht, Viglione & Blöschl, 2017; Dawson et al., 2011; Gilligan, Brady, Camp, Nay & Sengupta, 2015; Smith & Tobin, 1979) and the roles of individuals and groups in flood warning and evacuative scenarios (Haer, Botzen & Aerts, 2016; Haer, Botzen & de Moel, 2016; Nunes Correia et al., 1998). However, so far, little work has been conducted on whether evacuation strategies need to be tailored to the specific geographical setting or explored whether different modes of communication (direct or indirect) affect the evacuee's response. Answering these questions is important if effective warning strategies for specific places are to be developed.

More broadly, answering these two questions encompasses the process of implementing alternative actions; these rely on positive social participation, diffusion of ideas and their implementation, and they require broader acknowledgement of, and a specific approach to addressing, the associated socio-environmental complexity (Wisner et al., 1994; Wong & Luo 2005; Zarboutis & Marmaras, 2005). The HABM framework enables us to properly explore the systematic, cross-scale sensitivity of social complexity to the physical flood phenomena and shows where the loci of *vulnerability* are within an affected system. Therefore, the goal of HABM use for this study is not to eliminate complexity from consideration, but rather to harness it as a compliment to more specific physical considerations within comprehensive hazard management strategies. This is tested by applying it to a test case in Carlisle, UK. The overall aim is to offer an assessment of the value of alternative actions within flood hazard management as a whole (Dawson et al., 2011 and Müller, Bohn, Dreßler & Groeneveld, 2013).



2. **Methods**

2.1 **Study Area**

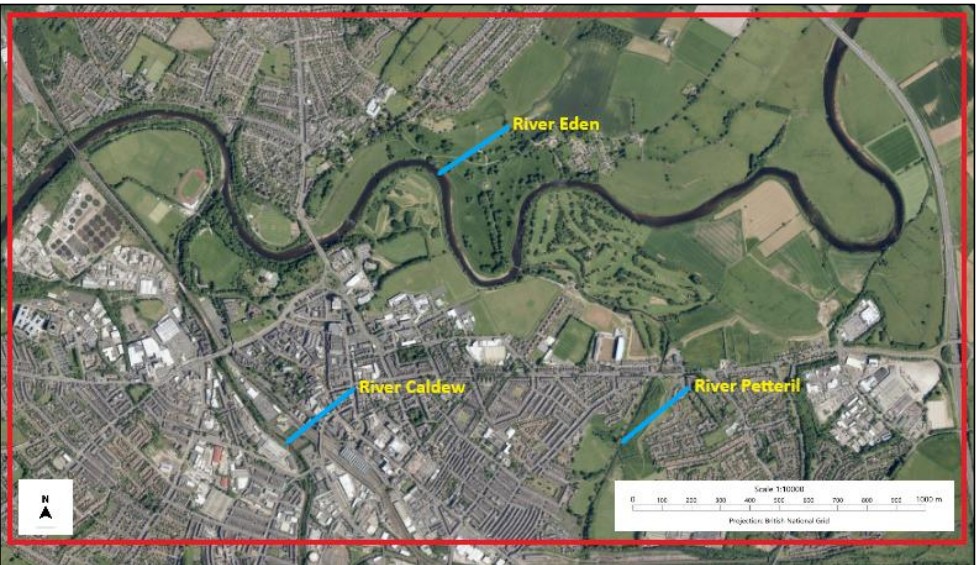

**Figure 1:** The area simulated in the HABM, highlighted in red with river locations indicated for the river(s)
Caldew, Petteril and Eden. (Contains OS data © Crown copyright and database right (2019))

Carlisle, Cumbria UK, and specifically the 10.3 km2 study area of the city illustrated in figure
1, is a flood prone city with a history of contemporary study (Correia et al., 1998; DEFRA 2007;
The Environment Agency, 2006, 2012 & 2016; Horrit, Bates, Fewtrell, Mason & Wilson, 2010;
Neal et al., 2009 and Neal, Keef, Bates, Bevan & Leedal, 2013).  Notable flood events have
affected the city since 1700, with the recent 2015 flood event having been referred to as
'unprecedented' in scale due to the river Eden's flood level rising 0.6 metres above the
previous record flood level of 2005. The location of the city at the confluence of the rivers
Eden, Caldew and Petteril means it is a useful source of data for hydrological research. As the
county town of Cumbria, with a total population of 108,000, it is a location of significant social
scale whilst also offering a case study which is suitably complex to develop new insights
through modelling and simulation.

The 2005 event affected approximately 1865 properties and led to the loss of 3 lives. The
event had an estimated Annual Exceedance Probability (AEP) of 0.59% (1 in 170-year return
period) and was a seminal event in that it prompted significant investment in the city's flood
defences. The 2005 LISFLOOD-FP data set (Horrit et al., 2010) provides a robust and reliable
foundation on which to build the agent-based component of the coupled model. This data set
used for the model simulation consists of a series of input files including raster grids of
floodplain friction coefficients and elevation heights in 2D, ARC-ascii format, boundary
identification, time-varying boundary conditions and hydrodynamics. Since 2005, Carlisle has
been subjected to further large flood events in 2009 and 2012 with the mitigative measures





deployed post-2005 successfully curtailing the impact of these. Furthermore, the 2015 event,
overtopped the new defences and has led the Environment Agency to produce the Cumbria
Flood Plan. A novel feature of this is that it introduces and promotes community-based flood
resilience measures on a large scale for the UK. It is the essence of these measures that
prompted the development of the coupled model with a view to better understanding the
dynamics on which these measures were based (DEFRA, 2007; Dugdale et al., 2009 and The
Environment Agency, 2006, 2012 & 2016).
2.2. **The flood modelling component: LISFLOOD-FP**
For a viable exploration of different individual responses to flooding, detailed, accurate and
dynamic simulations of the flood at Carlisle were required. LISFLOOD-FP (Bates & De Roo,
2000; Bates et al., 2010; Neal et al., 2009 & 2012), is a 2D hydrodynamic model specifically
designed to simulate floodplain inundation in an efficient manner over complex topography,
as is the case in urban areas. LISFLOOD-FP is capable of simulating grids of up to $10^7$ cells for
dynamic flood events with airborne laser altimetry defining the DEM of the affected area.
From this, the LISFLOOD-FP model can accurately simulate the dynamic propagation of flood
waves by predicting water depths in each grid cell through a series of time steps, and over
the complex topographic forms within floodplains. The ABM element of the coupled model
can then operate from this reliable foundation, enabling exploration of different hypotheses
for social reactions and responses to the detailed, accurate and dynamic physical outputs
generated by LISFLOOD-FP; by adding the related elements of policy and systematic change
(Wheater, 2006; Wilson & Atkinson, 2005). Whilst LISFLOOD-FP was the chosen hydraulic
model for the HABM, similar 2D-hydraulic models could resolve flow problems to similar
degrees of accuracy and this would mean that these alternative models could be utilised in
place of the LISFLOOD-FP with the HABM modelling framework (Hunter et al., 2008;
Landstrom, Whatmore & Lane, 2011; Neal et al., 2012).
2.3. **The social modelling components: HABM & NetLogo**
With LISFLOOD-FP producing an accurate representation of the flood at Carlisle, the related
elements of flood incident policy options and agent behaviour were implemented through
the separate ABM program of NetLogo (Railsback & Grimm, 2012 and Wilensky & Rand,
2015). The HABM (figures 3 to 7), uses water depth output files from the LISFLOOD-FP at each
model time-step within a simulated version of the affected area (figures 5 - 7). For the
simulation of the Carlisle study area, a Digital Elevation Model (DEM), identical to that used
by LISFLOOD-FP as an input data set was used to provide a realistic topography for the flood-
impacted area in NetLogo (NetLogo, 1999; Wilensky & Rand, 2015). In addition to the
simulation of the flood event and physical landscape, NetLogo was used to generate a virtual
population of *agents* to occupy the virtual version of Carlisle. Using a pseudo-random,
number of generator and deterministic agent scheduling algorithms directed through
probabilistic routines (Nunes Correia et al., 1998; Wilensky & Rand, 2015; Wong & Luo, 2005)
this then simulated the population's interaction with the environment and response to the
flood event. This simulated interaction allows the possibility of identifying *emergent*


*properties* likely to arise at the complex interface between the social and environmental
systems. These emergent properties have a significant impact on objective 1, in that they
occur subtly and at locations that significantly influence human responses within the coupled
physical and social systems. This significance is found in the HABM's capacity to reveal
systematic emergent phenomena through the simulated co-evolution of a socio-
environmental system, operating here through a flood event that has impact upon the basic
daily routine (figure 2) and the complex co-existent entities *i.e.* the more complex, responsive
*configuration* of evacuating groups  (figures 3 & 4). This then has a further impact on
hypotheses regarding *risk*, *vulnerability* and *resilience*, with the HABM providing an
opportunity to analyse and evaluate these terms, from a sub-systematic perspective. Here,
*sub-systematic* is a term used to describe the development of individual (micro) to community
(meso) level characteristics in response to the flood onset, with greater scope than has
previously been possible with traditional approaches to flood incident management
(Borschev & Filippov, 2004; Chen & Zhan, 2008; Gilbert & Troitzsch, 2005; Guo, Ren & Wang,
2008; Guyot & Holiden, 2006; Landstrom et al., 2011; Namatame & Chen, 2016, Sanders &
Sanders, 2004; Srbljinović & Škunca, 2003; Wei, Zhang & Fan, 2003)).
2.4. **The enhanced social modelling component: Bass Model**
For objective 2 of this paper, and in planning for effective flood impact management on a
broader scale, we must incorporate elements from a whole range of *activities* (Axelrod, 1970
& Berendracht et al., 2017). These include the spatial and temporal variations in phenomena
(flooding in this instance), the non-linear relationship between small perturbations at a sub-
systematic level and large knock-on effects at a system-wide scale ( the macro-level ), the
understanding that these effects can extend beyond the physical impacts of the phenomena
and change social behaviours and routines within an affected area, thus changing the
characteristic function of the system as a whole. This suggests that objectives 1 and 2 are
intimately connected and so there is a need to consider the social dynamics and reflexive
nature of the human system in response to the flood event within the framework of the
hazard system to determine the sensitivity of the incident management response (Davies,
1979). To better understand this relationship between human system and environmental
phenomena (figure 2), the ABM was used to provide choices to the simulated agent
population of Carlisle as part of a synthetic daily routine (figures 3 & 4), further details of
which are to be found in section 3 of this paper. These agent choices and the routine were
combined to synthesise the dynamics of the socio-environmental interface and from this,
estimates were made for the influence that agent choices have on the characteristics of the
system being simulated.  In the Carlisle HABM, the agents were given the choice of carrying
out their normal, linear, routine during the flood scenario, of becoming warned and taking
immediate action to evacuate, or of assessing this warning based on social interaction with
other agents in the immediate vicinity, and then acting post-interaction(figure  4). The
scenario of becoming warned and *evacuating immediately* is used in the HABM to reflect the
government policy instruction of 'what to do in a flood scenario' in the most direct form.
Within the model (DEFRA, 2007), this instruction is programmed as *'pre-preparedness'* and it
describes an adoption and undertaking of actions beyond the 'normal' daily routine, both
modelled and real (Chen & Zhan, 2008; Chu, 2015).
The Bass Diffusion Model provides a tool for interpreting the impact of these choices and
actions, by representing agents who adopt certain actions at a given time. The model,
originally conceived for marketing economics, is used to inform understanding of the diffusion
of frequently purchased or *adopted* products, and is based on a principle derived from the
following relationship (Bass, 1969):
$$\frac{f(t)}{1 - F(t)} = p + \frac{q}{M}\left[A(t)\right]$$
This states that "The portion of the potential market that adopts at time *t*, given that they
have not yet adopted, is equal to a linear function of previous adopters" (Bass, 1969; Davies,
1979). The basic premise of the model provides insight into interaction between adopters of
the *product* within a population; it then classifies these adopters as '*innovators*' or '*imitators*'.
In the HABM, the 'material product' concept of the Bass Model is replaced with the apriori
product of 'knowledge' regarding an imminent flood event, this is to say that agents within
the model can simply be set to act out evacuative measures immediately at the start of the
simulation and in all of the timesteps leading up to the flood inundation, if they choose to
stay. These 'innovative' agents are also freely able to communicate these measures to
proximal neighbouring agents who can then choose to imitate these informed agents; or carry
on with what they are doing. it should be stated that the sociological dynamic of *innovation*
and *imitation* is proliferated within the model by communication between agents who are
proximal and so this simple binary distinction could be regarded as a potentially useful one
for representing the apparently complex communication dynamics of a social system in a
relatively simple manner.
In the specific instance of the HABM, the *innovators* are set as *pre-prepared* prior to the flood
simulation onset and the *imitators* are those who would not be prepared, but who are given
the choice to adapt their routine at each timestep, based upon contact with the innovators.
This situation, describing people who are in possession of knowledge regarding the flood
event and then communicating it to those who are not, could have an impact on all aspects
of response and evacuation, as it is a crucial component of the boundary between the
processes of *warning* and *response* (Axelrod, 1970; Chen & Zhan, 2008, Chu, 2015). With
specific reference to the Bass Model terminology, there are three parameters (or
representative coefficients), that define the compatibility with the HABM, these are:
• (*M*) - The potential *market*, these are the ultimate number of potential adopters, i.e. the
population. This constitutes the number of members of the social system in which word-
of-mouth communication from past adopters is the driver of new adoptions. The Bass
Model assumes that *M* is constant, though in practice and over longer periods, M is often
slowly changing according to population change and product memory.
• (*p*) – The coefficient of innovation, so-called because its contribution to new adoptions
does not depend on the number of prior adoptions. Since these adoptions are due to





some influence outside the social system, the parameter is also called the "parameter of
external influence.'
• **($q$)** – The coefficient of imitation has an effect that is proportional to cumulative adoptions
A(t), implying that the number of adoptions at time t is proportional to the number of
prior adopters. In other words, the more that people talk about a product, the more other
people in the social system will adopt it. This parameter is also referred to as the
'parameter of internal influence'.
The other variables in the Bass Model relationship and calculated from $M$, $p$, $q$ and $t$, are:
• *f(t)* - The portion of M that adopts at time t,
• *F(t)* - The portion of M that have adopted by time t,
• *a(t)* - The adopters (or adoptions) at t,
• *A(t)* - The cumulative adopters (or adoptions) at t.
The outcomes of the coupled application of these three components (sections 2.1, 2.2 & 2.3)
towards the two objectives are further illustrated in section **4** and are discussed further in
section **5**.
Of further interest here is how to qualify the communication taking place within the HABM.
In sociological terms, the imitative process involved is broadly one of inter-agent
communication and collective response. According to the sociologist Gabriel Tarde and his
Laws of Imitation (Tarde, 1903), as applied to 'groups of people', innovations must undergo a
process of diffusion over time to gain a foothold and become a component in the decision-
making process linked to the innovation, be this *adoption* or *rejection*. Tarde's process
involved in the diffusion of innovation has undergone some revisions in the decades since
being first proposed and can now be defined through the following five steps:
• First Knowledge,
• Attitude formation,
• Adoption or rejection,
• Implementation,
• Confirmation of the decision.
Via the Bass Model, the HABM for Carlisle allows a simulated engagement with the first four
steps of Tarde's process, the fifth being confirmed in the representation of the first four
activities as the simulation advances over time. This interpretation of social imitation and
adoption was used as a basis for investigating the influence of these processes in an event
where time is relatively constrained and the stakes of action are high, such as during a flood
onset. The values for this process of adoption were taken from the change in overall un-
prepared population in Carlisle transitioning to a 'prepared state' based upon contact with a
'pre-prepared', or innovative, agent. This transition was represented by the percentage of the
population in possession of the appropriate knowledge for effective flood evacuation who



then reported this change back as an agent-orientated change of state throughout the
simulation of the flood. This rate of change of state is then fed into the Bass Model functions
to produce diffusion curves like those seen in figures 8a & b and discussed in further detail in
sections 4 and 5.
3. **Core model construction and system dynamics**
Given the complexity caused by the incorporation of these diverse elements within
considerations of a flood hazard system, the benefits of a standardised flood incident
management strategy based on an understanding of these dynamics might not be
immediately apparent. Further management of complexity might necessarily arise through
the required interactions between the individuals and organisations who might very well have
conflicting interests linked to contrasting elements in their expertise or experience (Hart,
Nilsson & Raphael, 1968; Hornor, 1998). Furthermore, the feedbacks within a flood hazard
system, particularly an urban one, can lead to a spectrum of dampening and amplification of
behaviours within the system, the dynamics of which could be influential on outcome, yet
difficult to account for in a standardised flood incident management strategy (Assaf &
Hartford, 2002; Dawson et al., 2011, Rasmussen, Pejtersen and Goodstein, 1994.) . It is here
where the HABM concept reaches out to the concepts of phenomenology, poststructuralism,
structuration theory, structural functionalism and symbolic interactionism to inform the
conception of a modelling framework that incorporates the important social notions of these
disciplines and thus anchors the modelling element of the HABM to the cardinal philosophical
and sociological concepts underlying it and the outputs produced. The appeal of this approach
lies primarily in the novelty of the undertaking in addition to the application of concepts from
disciplines such as sociology, philosophy and psychology, which complement the model by
offering access to new terminology and theoretical bases for better representing social
systems, focussed on *relatedness* rather than *boundedness* between the dimensions and the
whole (Alexander, 1980) ; within a coupled modelling framework. Here, the benefit of a more
holistic representation can lead to the development of a more effective and holistic
understanding of how to manage social dynamics, responses and functions within physical
models where they can have further impact on effective planning for and outcomes from the
whole system and the components comprising that system (Smith & Tobin, 1979; Zarboutis
& Marmaras, 2005).
With these details in mind, and urban systems being the primary interest in this paper (figure
2), the first step beyond bringing together the initial HABM components was to devise a
conceptual format that describes the key dimensions of the urban system within a
parameterised and reproducible framework. In this paper they will be primarily referred to as
*dimensions*, alternatively they can be called '*sets*' (or *centres* (Alexander, 1980), and can be
broadly subdivided into three separate systems, that of the *Environment*, *Community* and
*Built Infrastructure* (UNISDR, 2015; Wisner et al., 1994). Networks existing between these
dimensions, resulting from the co-evolution of the dimensions, are characterised by the
immediate practical and physical influence that each has on the behaviour of the other to
create an operational whole. Conceptually, this is analogous to the notion of the *Brunnian*
*Link* in mathematics and the poststructural, psychoanalytical concept for *experience* or
*jouissance*, proposed by Jacques Lacan's *Borromean Rings* construct in the 1970's (Zupančič,
2000). An urban system, concomitant with our physical perception and experience of it, can
occur at the nexus of the topological sets illustrated in figure 2. Whilst these constituent
dimensions could be deliberated in terms of scale, dynamic or boundary and seemingly
experienced separately from one another by individuals or groups, it is important to
understand that for the present analysis, the function of the urban system within the HABM
framework arises in the form of the aforementioned Brunnian link. This is as an "extended
and unbroken continuum of connections wherein the whole is necessarily unbroken and
undivided" so that life may be supported, experienced and proliferated therein (Alexander,
431  1980).

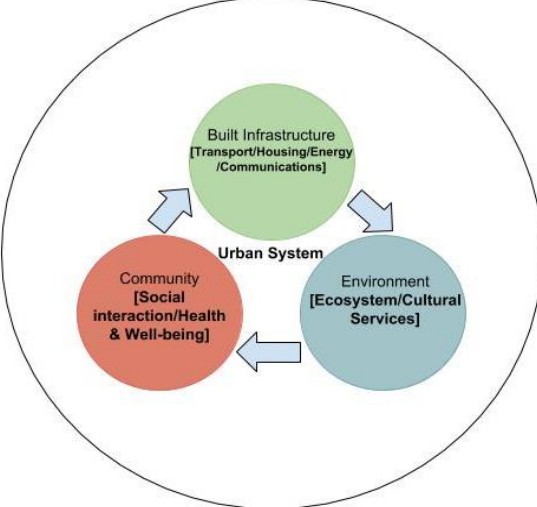

**Figure 2:** A simplified schematic illustrating the key centres of an urban system. Conceptualised from [Axelrod, 1970, Wisner et al., 1994] and the terminology given within the Sendai framework 2015-2030 [UNISDR, 2015].

Specifically, this link is a mathematical and topological term used to describe the triviality and
non-triviality of connection between the sets. As applied to the HABM system concept, when
disconnected from the complete, interconnected, system set, the system no longer exists and
cannot be experienced by people within it. Utilising the terminology applied within
mathematical topology, the individual systems become '*trivial*' when disconnected from one
another and '*non-trivial*' when all are in contact within the dimensions of the systematic
whole. Thus, the individual systems are experienced in combination with one another, where
the boundaries, existing between these systems, would not be as discrete as those shown in
figure 2. This would suggest an overlap in the systems whereby experience and interactions
between these systems and people, *life*, occurs at the nexus of the three. A simplified scenario
to support this understanding for Carlisle would be one where a *community* requirement for
an advance in *built infrastructure* as a response to perceived, or experienced, *environmental*
risk from flooding; something which could be considered an *emergent* characteristic from the





onset of the flood hazard system. Consequently, were the topologies of each of the three
dimensions existent separately, and not connected in a manner as suggested in figure 2,
interactions between the elements of the three system sets, including the manifestation of
physically *hazardous* phenomena, would not be possible (Alexander, 1980; Axelrod, 1970;
Berendracht et al., 2017; Du et al., 2017; Dugdale at al., 2009; Eberlen et al., 2017; Fordham,
1992; Guyot & Honiden, 2006; Holland, 2014; Liu et al., 2015; UNISDR, 2015).
Thus, the simulations of the dynamics of Carlisle's urban system for the HABM focused on
establishing the linked characteristics between the three dimensions to model a non-trivial
system. The use of an ABM enables this through a focus on the community dimension,
through simulation of activities and interactions which may then be used as metrics for
change according to a specific environmental event, in this instance the 2005 flooding of the
Rivers Eden, Petteril and Caldew. To perform these simulations, a correspondence between
the conceptualised urban system, representing the three inter-linked elements of figure 2 and
the modelling framework illustrated in figure 3, was developed. Figure 3 is a schematic of this
correspondence and represents the overlying workflow of the HABM for simulations of the
2005 Carlisle flood. The layout for this figure was used to support workflow and model
structure in relation to effective representation of the urban system shown in figure 2, within
the ABM platform. The Layout of figure 3 is such that the structure of each set from figure 2
corresponds with the processes taking place in NetLogo to represent that set. In sum:
• The environmental set is simulated using the LISFLOOD-FP outputs and the site DEM,
• The Built infrastructure is emulated using census data sets and street network
information
• The community or social set overlaps both the built and environmental systems and
is driven by the agent-orientated, probabilistic choice and interaction flowchart
illustrated in figure 4**.**
The details of the diagram in figure 3 are the cardinal NetLogo commands that overlap
between the system sets and so enable the simulation of the three dimensions within the
HABM. This establishes a tangible link between the conceptual complexity of the urban
system experienced by people with that experienced by agents, who represent people, within
the simulated version of the urban system. This transferral from a conceptual topological
figure to a logical modelling schematic was an important step which was taken to link the
modelling system to the physical system being modelled. Whilst the format presented in
figure 3 is not particularly novel in the sense of workflow or process for an ABM, it is relatively
novel in the sense of how it illustrates this link between a conceptual construct of a system,
figure 2, and the workflow steps required in simulating this system and representing dynamics
that can provide an analogue for events that occurred during an historical physical event, such
as that in Carlisle during 2005.


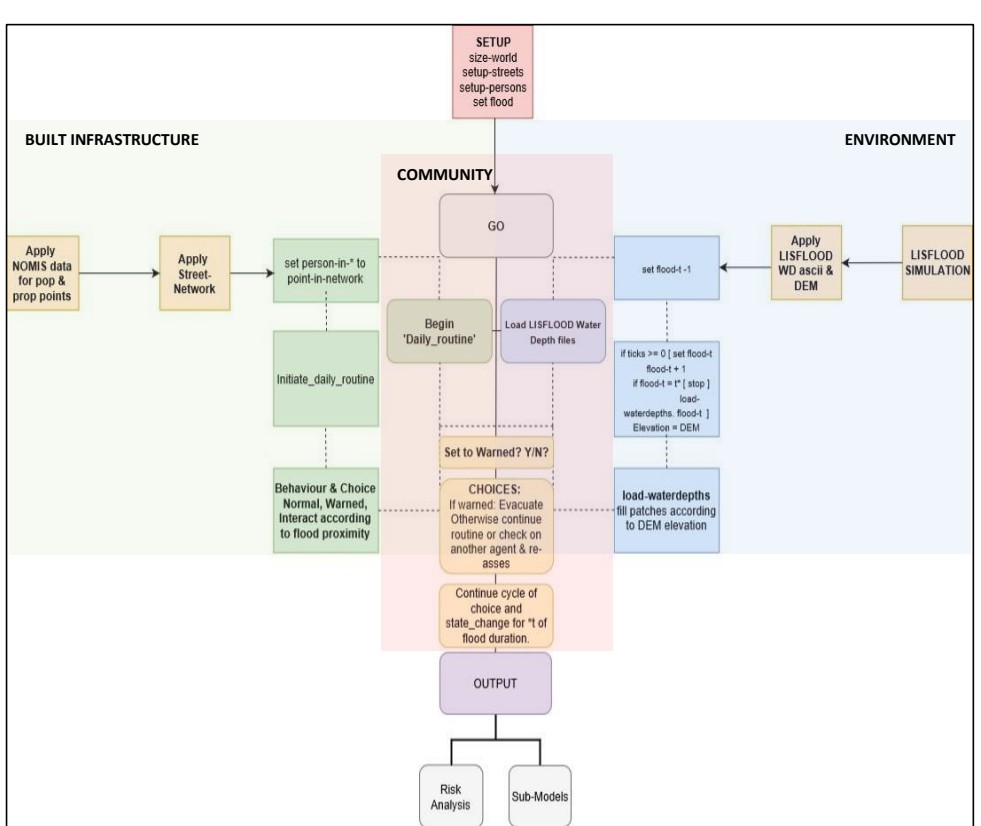

**Figure 3:** The core components of the HABM, an indication of the model cycle for these components, and the elements of the urban system (**figures 1 and 2**) that they demonstrate. The schematic follows a similar format to that of a Euler diagram [Whitehead & Russell, 1913], whereby the three centres of the urban system are shown to contain the respective components of the model representing their function within the HABM. These are (from right to left): Built Infrastructure, Community, Environment.

Figure 4 further extends this conceptual approach through to the community element of the modelled system in offering simulated agents the choice to engage with a basic, probabilistic, daily routine within the simulated system as well as engage in emergency response actions following flood onset. This further enhances the realism of the simulated population of Carlisle and provides an analogue for how variations in the physical interaction with a flood might affect the evacuation response (Morss et al., 2016; Müller et al., 2013). The routine and decision tree format, formulated through the ODD (Overview Design concepts & Details) protocol (Wilensky & Rand, 2015), with a view to potentially producing '*emergent*' behaviour for the modelled system, was initially referenced from the synthetic daily routine and transport model used for simulating storm-surge evacuation by Dawson (et al., 2011). The adopted elements of this routine were the basic formatting seen in figure 4, whereby probabilities were assigned to activities for the agents in the model. These activities were engaged with on a point-to-point basis as the agents navigated through the simulated system of Carlisle until flood onset. With onset, the agents within the simulated system can then

choose to engage with the emergency routine or continue with the elements of a daily routine
until the next timestep. As there is already a wealth of evidence available (see for example:
Assaf & Hartford, 2002; Berendracht et al., 2017; Chu, 2015; Du et al., 2017; Dugdale et al.,
2009; Eberlen et al., 2017) to suggest that the time of event onset is influential in event
outcome, this time-dependency was not implemented within the simulations for Carlisle. This
choice was made in favour of developing streamlined simulations that emphasised agent-
agent interactions between event onset and end. However, time-dependency is something
which is easily implemented within NetLogo if desired and indeed was implemented in later
iterations of the HABM for different applications. In addition to this agent-agent focus, non
'pre-prepared' agents may also engage with 'pre-prepared' agents in the model and initiate
emergency action based upon their interaction, demonstrating a synthesised form of
communication and response. The development of this step in the modelling procedure was
crucial to allow interpretation of the influence of an adopted policy directive on inter-agent
interaction and choices made during the onset of the flood event which may ultimately not
be time-dependent in nature (DEFRA, 2007;Landstrom et al., 2011; Liu et al., 2015; Morss et
al., 2016; UNISDR, 2015; Waldorp, 1993).

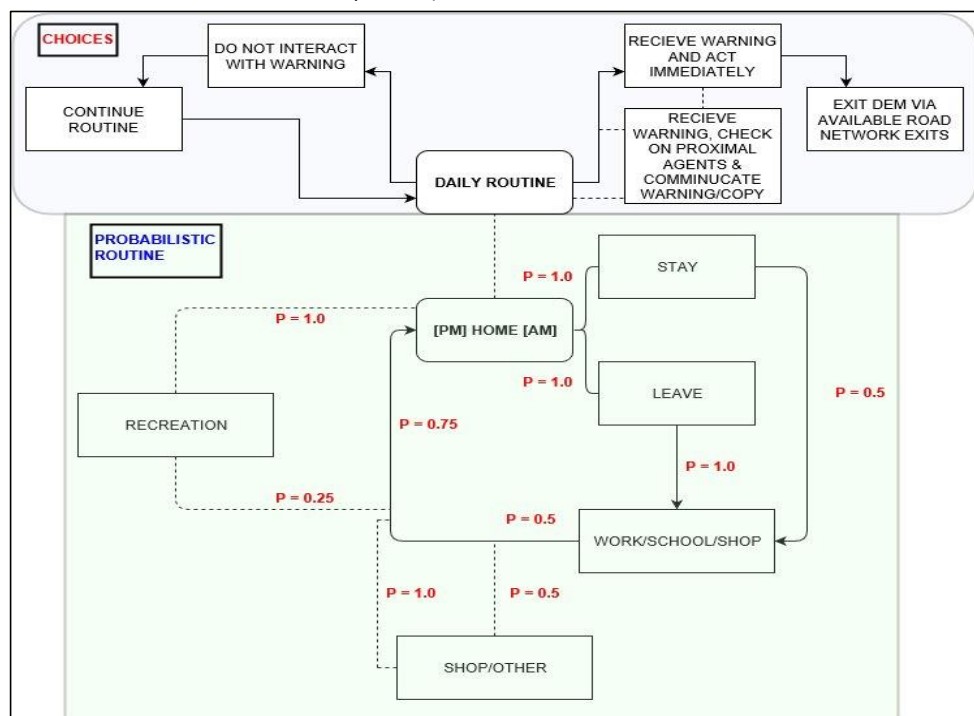

**Figure 4:** An overview of the agent choice & probabilistic routine tree used to guide agent processes through
the simulated environment of Carlisle. Informed by reference to (Bennet & Tang, 2017) & (Dawson et al., 2011).
The format of figure 4 was beneficial in this instance as it offers a basic format for agents
operating within the model of Carlisle, a format by which they can navigate along the street
network in a manner reflective of what might be expected during an average day in Carlisle.
The probabilistic format of the routine ensures that upon each timestep agents will be at
specific points within the network. Whilst this attenuates the representative complexity of
the model, it is believed that it offers enough complexity of choice and action to reflect the
potential reality of a complex social and flood onset situation within Carlisle. The probabilities
shown in figure 4 were adapted slightly from the original synthetic routine proposed by
Dawson et al. to be more generalised and, for computational efficiency within NetLogo, were
implemented to be acted out on each time step, rather than continuously over flood onset.
In section 4, figures 5 to 7, the product of the co-action between the components of figures
2, 3 & 4 can be seen. These figures illustrate the model in a preliminary state of simulation
and so the full agent population is not in action. Whilst the largely autonomous processes of
NetLogo, outlined in section 2, influenced the extent to which the simulated agents engaged
with the routine and the choices provided, the implementation of a routine acted to
attenuate not only the representative complexity of the situation, but the outright
stochasticity of the NetLogo agents also. This means that whilst the agents would be
interacting with 'commands' *e.g.* 'leave home point' or 'stay at home point for t(n)', these
commands are not too far removed from a realistic analogue of basic choices a human might
make on a given day (Chu, 2015; Dawson et al., 2011) with the possible actions of the daily
and emergency routines being more reflective of general and reactive behaviours expected
during a flood onset (Du at al., 2017; Dugdale et al., 2009). The spatial distribution of the agent
population within the HABM was informed with national UK Census statistics for Carlisle.
However, as census data does not identify individuals against specific addresses, the
distribution of agents within the simulated HABM environment was implemented in a slightly
more utilitarian manner than the demographic-based distribution seen in Dawson (et al.,
2011), by  using a linear function of the population of Carlisle with agents being allocated to
home points within the model according to building footprint (Bennet & Tang, 2017; Borschev
& Filippov, 2004; Dechter & Pearl, 1986).
In terms of the Bass Model variables discussed earlier, (**M**) is represented by 108,000 agents
(in the final simulations), the total population of Carlisle (The Environment Agency, 2016); (**p**),
here, represents the 50% estimate by the EA for the population of Carlisle currently deemed
as 'signed up to flood warnings' or *pre-prepared*  and in possession of the innovative
knowledge to respond to the flood upon onset (The Environment Agency, 2012). The
coefficient (**q**) roughly equates to 30% which represents the one-third likelihood of those who
encounter the innovators (**p**) adopting the innovation as defined by the Bass Model in a
scenario where the rate of adoption between innovation and adaptation is linear or *seamless*
(Bass, 1969).   Despite this somewhat ideological perception of human communication
(Jakkola, 1996), this rate of conversion was kept consistent in the instance of the Carlisle
simulations as no evidence was found to suggest that social factors were present within
Carlisle that would adversely affect it (widespread prejudice, social unrest, a despotic
government etc.) In total 200,000 simulations were performed using this methodology within
the NetLogo BehaviourSpace tool. These differed through scaling of 'pre-preparedness'
between 0 and 100% and the outputs of interest from these simulations were the rate of
change from '*un-prepared*' to an '*evacuative*' state, based upon agent contact and the
number of potential casualties linked to the change of preparedness (%). Finally, regarding
the status of 'potential casualties' within the HABM, this is a term and metric of the HABM
used to describe agents physically impacted by the flood. This term does not account explicitly
for 'death', rather it is a measure of those agents who may become cut-off from a clear escape
route or inundated during evacuative procedure and actual agent fatality was extremely rare
during the simulations. The simulation of fatality was defined differently to physical fatality
in that it was only presented when an agent's grid cell became inundated, to a third of an
agent's height, for one time-step, having had all escape routes cut off (Assaf & Hartford, 2002;
Landstrom et al., 2011; Roland & Moriarty, 1990).
4. **Results**
Figures 5 to 7 are examples of these simulated flood sequences for the 2005 Carlisle flood by
the HABM, showing inundation areas and agent locations, both prior to the flood (figure 5)
and at later stages (figures 6 and 7) after flood onset and agents have been variously alerted.
Side panels on the left-hand side of the figures outline the basic controls for the model, whilst
the charts on the right show model predictions for *potential casualties* in relation to
populations and *pre-preparedness*, which is an apriori knowledge of the flood, as previously
stated. These figures are representations of the modelled culmination of the concepts
discussed in sections 1, 2 and 3 and illustrated in figures 2 to 4 within the NetLogo interface.

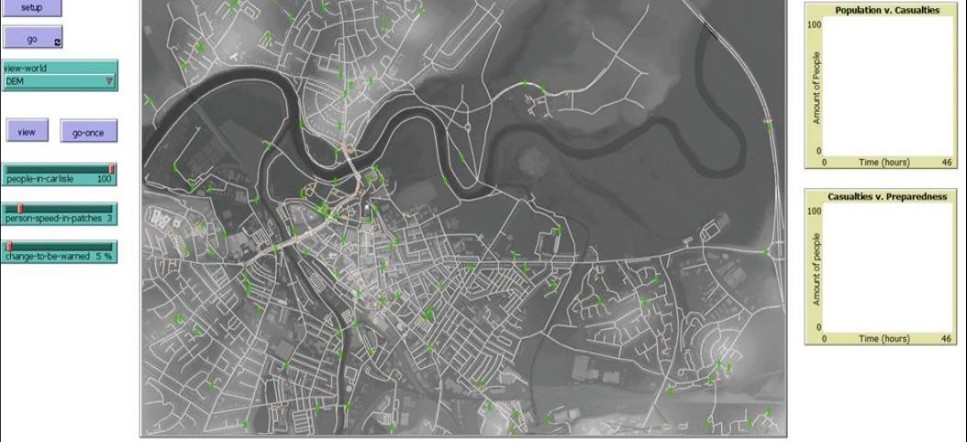

**Figure 5:** An overview of the preliminary HABM. Shown here as an example are agents engaging in the daily
routine (green) prior to the initiation of the LISFLOOD-FP flood inundation. These figures represent only a small
proportion (<1000 agents) of the full agent populations (~ 108,000 agents) simulated for the final results of the
simulations


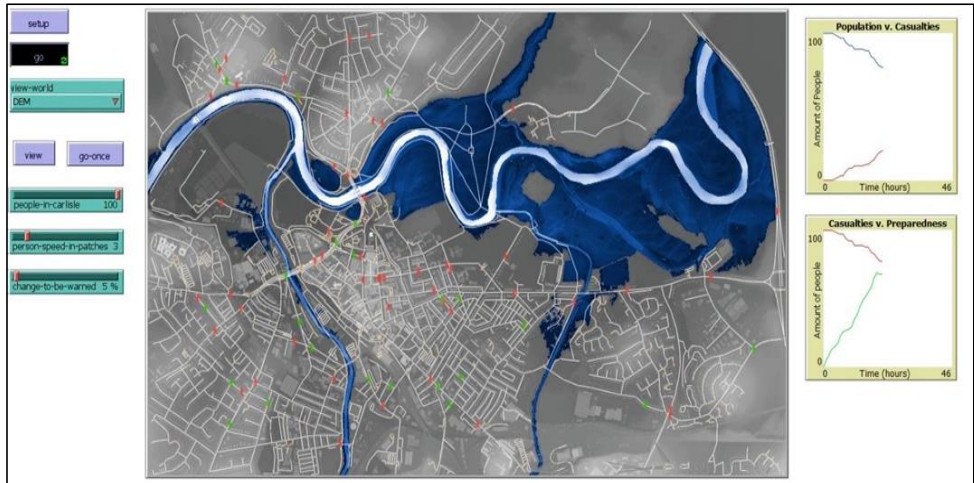

**Figure 6:** Agents marked in red are those whom have become aware of the incoming flood and are taking evacuative action. Changes in agent colour on the GUI (Graphic User Interface) indicate that members of the sample population are transitioning to a 'potential casualty' as the flood encroaches their vicinity but also that the likelihood of casualty occurring will diminish over time as the message of 'preparedness' diffuses through the population.

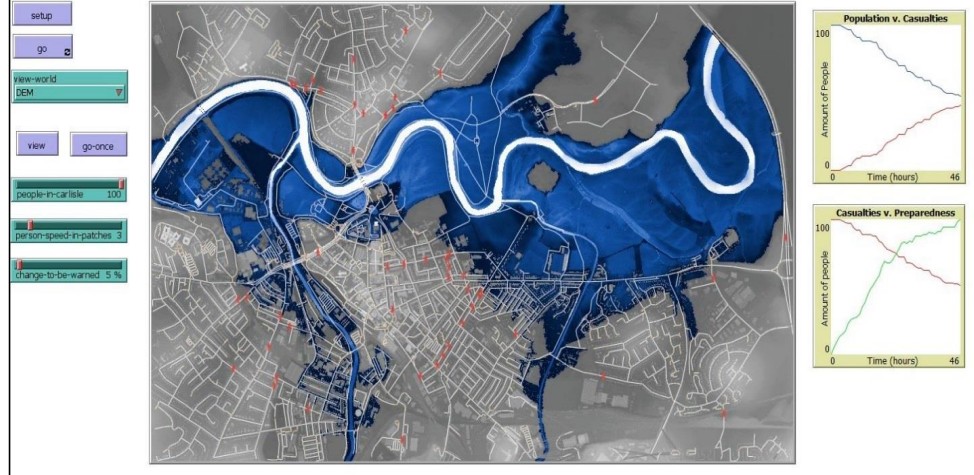

**Figure 7:** Further to preparedness and potential casualty, an indication of areas in which people are likely to stay, areas from which people are most likely to move as well as the areas through which people are most likely to pass are shown within the HABM GUI.

In applying the Bass model to the Carlisle HABM, two diffusion curves were produced (figures 8 a & b). These represent inter-agent communication regarding the adoption of policy instructions to either evacuate the area immediately, i.e. to adopt an innovative instruction, or to follow an imitative one after checking with nearby agents and only then deciding how to respond. The coefficient (**q**) is typically represented by a much smaller value than 30% in traditional applications of the model (Mahajan, Muller & Bass, 1990). However, owing to the


elevated risk involved in adopting, or not adopting, the product of evacuative knowledge
during a hazard scenario, the traditionally small value of (**q**) has been scaled up significantly.
This is to represent a one-third likelihood (~30%) of those who encounter the innovator (**p**)
agents, receiving the flood warning by communication and adopting directly from them.
Whilst this is a manipulation of the Bass Model function, it remains consistent with the Bass
Model theory, stipulating that human adoption of a process or product is more likely to
happen based upon internal systematic influence, or *imitation*, rather than through external
influence on the social system, or by *innovation.* Wherein the available choices may be
reduced to 'yes', 'no' and 'maybe', probabilistically represented as roughly one-third each for
a given scenario (Dechter & Pearl, 1986; Hart et al., 1968; Hornor, 1998; Mahajan et al., 1990;
Massiani & Gohs, 2015; Sultan, Farley & Lehmann, 1996).
The fundamental difference between (**p**) and (**q**) is generated from this external-internal
distinction. Aligning this further with the sociological notions of Tarde, (**p**) is a representation
of an external factor that requires a change in operation of the internal system dynamics (**q**)
over time, thought of as an attunement, harmonisation or, in more traditional terms; an
*acceptance* (Tarde, 1903). This means that for an innovative process (**p**) to become a
naturalised component of the internal system dynamics (**q**), a significant amount of time may
be required for innovation to lead to imitation when there is a *risk* involved (**63**). In this
application, the Bass model gives an indication of this duration based on the relative
probabilistic magnitudes of (**p**) and (**q**) for a population of 108,000 agents. The overall
significance of this application is that it allows conclusions to be made as to how influential
external policy protocols are for the population in relation to their internal 'sense' during
flood event response (Massiani & Gohs, 2015, Sultan et al., 2003).
The curves illustrated in figures 8a and b are the separate curves for the process of adoption
based upon the optimised Bass Model values for the coefficient of innovation (**p**) at 50% and
coefficient of imitation (**q**) at approximately 30% over 200,000 simulations for the Carlisle
model. The three separate lines are illustrations of the three different iterations of the
model's standard differential equation as functions of continuous and discrete time (**5**).
Correspondence between the curves represents an agreement between the model's
functions and the data being plotted. Broadly, the curves show that the external directive,
seen in figure 8a (**p**) is more effective at promoting an immediate evacuation as a lower
number of the simulated population changing states over time would suggest that a large
proportion of the original innovators choose to act in the early onset of the flood and
evacuate the area without hesitation. The negative aspect of this function is that there will be
less agents available to communicate the innovative process of (**q**) and influence the less
prepared agents and so this process of innovation will take longer to diffuse throughout the
agent population leading to less agents taking appropriate action and exposing themselves to
potential danger.
The curve for figure 8b, (**q**), is the internal function for evacuative measures, which is reliant
on agent-agent interaction and suggests that the internal dynamics for the adoption of
evacuative measures, that is to say the adoption of the same actions as the agency directive
but not directly from the external directive (e-mail, text alerts etc.), according to
communication between agents, within the total flood affected population of Carlisle, is more
influential over a shorter duration than the operation of (**p**). The variance between the three
lines would suggest that there is some disagreement between the baseline functions of the
Bass Model differential equation and those for discrete and continuous time for (**q**) and it is
believed that this is likely related to the unusually high value attributed to the 30% likelihood
of agents agreeing to imitate the innovative agents and become imitators, as well as the
general stochasticity related to the reliance on 'proximal contact' for communication
between agents, which is likely but not guaranteed in any situation; particularly in one as
frenetic as that involving a flood.

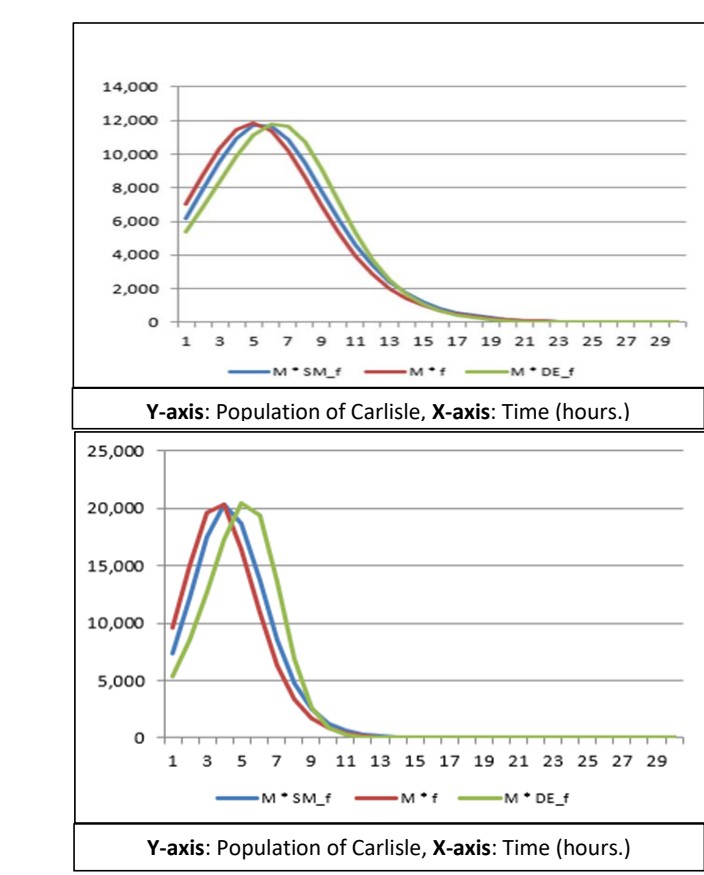

**Y-axis**: Population of Carlisle, **X-axis**: Time (hours.)

**Y-axis**: Population of Carlisle, **X-axis**: Time (hours.)

**Figures 8 a & b:** Example Bass diffusion curves for p (top) and q (bottom) at Carlisle. Illustrated are the curves
for the continuous time Bass Model functions (blue/ M*SM_f & red/ M*f) for discrete and incremental time-
steps and the Bass Model differential equation (green/ M*DE-f). The Y-axis for both curves
This bridge between sociological and theoretical concepts of process diffusion, or between
internal and external components, provides insight into the relationship existent between
policy and responsive behaviour. Furthermore, the Bass Model's use in the analysis of flood




response dynamics is a broadly useful one, providing quantitative evidence of behaviour, in
the form of diffusion curves (figures 8a & b) and, for the dynamics of during-event agent
communication, thus implementing Tarde's sociological laws into the modelling process. In
addition, it represents both the 'innovative' *i.e.* individual response to policy direction, and
the 'imitative' processes related to this direction, which certainly have influence on the micro,
and potentially macro, scale human responses to flood events (Guyot & Honiden, 2006).
As the flood depths in the Carlisle dataset were relatively shallow beyond the river channel
during the early time-steps, very few agents were presented with a potentially fatal scenario
that they could not escape from, registering them as a '*potential casualty*' instead of a fatality.
Broadly, a *fatal* scenario in this instance was determined by total cell inundation surrounding
an agent and preventing them from leaving. Whilst there are examples of models utilising
depth and velocity as determinants for a fatal scenario (Chen & Zhan, 2008; Chu, 2015;
Dawson et al., 2011) these were not functions implemented in this preliminary model but
were implemented in the later iterations of the HABM. Whilst the HABM should not be
regarded as a full predictive tool, it does enable the visualisation of individual and group
interactions, which might lead to potential casualty over repeated simulations. This is a
valuable insight given that it is often difficult to identify comparable levels of detail from
historical examples and their related data for micro-scale factors that are influential in event
outcome.
According to figure 9, once overall 'preparedness' of the agent population of Carlisle exceeds
30%, either through increased social interaction or directly from policy instruction, the
likelihood of 'casualty' resulting from the flood scenario actually increases. This was an
unexpected outcome and might, at first, seem counter-intuitive but is thought to be
attributable to the urban '*fabric*' (topography and morphology) of Carlisle. When agents
select to respond to the flood collectively and all at the same time, congestion of exit routes
leads to an overall reduction in of movement away from flood inundated areas, so increasing
agent exposure to the hazard (Wei et al., 2003, Werrity et al., 2007). This possibility is a
valuable new insight produced by the HABM. Figure 9 illustrates the range of results from the
200,000 simulations of the 2005 Carlisle flood. Across these simulations, the percentage of
the population pre-warned of the flood event was varied between 10 and 100 %. The current
DEFRA estimation for Carlisle is that 50% of the population (~ 54,000 people) are classed as
'prepared' for a flood (termed 'population warned' or 'pre-prepared' in the HABM
simulations.) The population warned within the HABM will initiate evacuative behaviours,
according to policy instruction, within the first hours (~ 1-3 timesteps) of the flood inundation
taking place and are able to communicate this action to surrounding agents from the outset
of the simulation, largely by-passing the time required for the autonomous decision-making
process during the event and engaging directly with the apparent agent preference for
imitative behaviour.


**Total number of potential casualties vs. % of population pre-warned for Carlisle.**



**Figure 9:** A box chart illustrating the range of values, sampled from 1000 agents (the most computationally stable
sample size for batch runs on the available architecture) within the full agent population (108, 000), for the total
number of potential casualties vs. % of population pre-warned for Carlisle from 200,000 simulations.

To assume that a higher percentage of pre-prepared agents would lead to an overall reduction
in potential casualties would be a logical assumption to make (Axelrod, 1970; Chen & Zhan,
2008; Dawson et al., 2011; The Environment Agency, 2016). As highlighted by figures 9 and
11, overall potential casualties for the simulated population of Carlisle shows an *increasing*
trend for higher percentages of pre-warned agents, particularly above 80% preparedness. As
already mentioned, this reflects the way in which Carlisle has been constructed around the
confluence of the river(s) Eden, Petterill and Caldew. It highlights the deficiencies of this urban
structure when a large inundation event forces significant numbers of agents to evacuate
through a limited number of escape routes (figure 10), (Gilligan et al., 2015; Sanders &
Sanders, 2004).

According to the HABM results, Carlisle's agent population has a distinct 'preference' for
evacuation to the south-west of the city, along the arterial A595. This preference was
established through visual assessment of the simulations and was likely determined by the
number of sub-routes that had access to the A595 and that were not cut-off by flood waters.
Indeed, the most densely populated areas of Carlisle are divided into four distinct areas by
the three rivers shown in figure 1 and so this preferred escape route is only immediately
available to those who are either pre-prepared, reside within the immediate vicinity of the
A595, or who live or work to the west of the Eden and Caldew. As the flood progresses beyond
the first 5-6 hours of propagation, the number of escape routes diminishes yet the number of
agents prepared to evacuate has increased significantly. This creates a backlog in the system
whereby more agents choose to stay in their immediate vicinity or to evacuate at the same
time as everyone else, exacerbating the system congestion and increasing agent exposure to
the flood inundation. Whilst agent choices do vary from simulation to simulation according
the choices of their routine and the type of agents they make contact with, this pattern of
evacuation occurs across the whole set of simulations, and so could be taken as an indicator
of likely choices made by the population of Carlisle if a flood happened today.

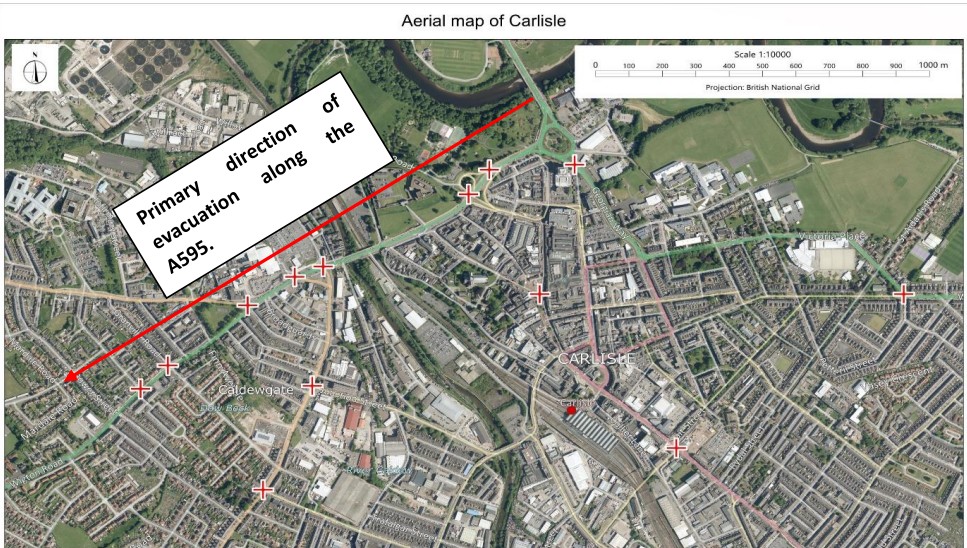

**Figure 10:** An aerial image of Carlisle illustrating the preferential direction for escape to the south west along
the A595. Further illustrated are the most prominent chokepoints (**red crosses**) for reduced evacuative flow of
people between 80 and 100% preparedness. These points were identified from the HABM as the nodes in the
street network overlay which have the most consistently high densities of agents throughout the range of
simulations. (Contains OS data © Crown copyright and database right (2019))

As is illustrated in figure 11, with less than 30% preparedness, agents within the HABM show
a preference for evacuation away from Carlisle during the earlier stages of the flood onset
and so the social response to the flood is slow when there are fewer people in Carlisle to
disseminate the message of evacuation. This finding further reinforces the results presented
in the diffusion model (figures 8 a & b). Without a threshold number of the population being
aware of the impending flood there is less likelihood of contact with unaware agents. This
means that the response dynamics are more reliant on the innovative procedures of policy
uptake and arbitrary choice, both of which are shown to be less likely to produce a *successful*
evacuation outcome. The transition from micro to macro level response, from individual
agent interaction up to a large group response to changes in the environment, is realistically
a much more complex process than that illustrated in the HABM model. Thus, as a starting
point for testing hypotheses related to transitory-scale flood hazard response, it is a useful
tool for exploring the related and inherent complexity of the socio-environmental interface
present during a flood event (Wilensky & Rand, 2015; Wisner et al., 1994; Wong & Luo, 2005).

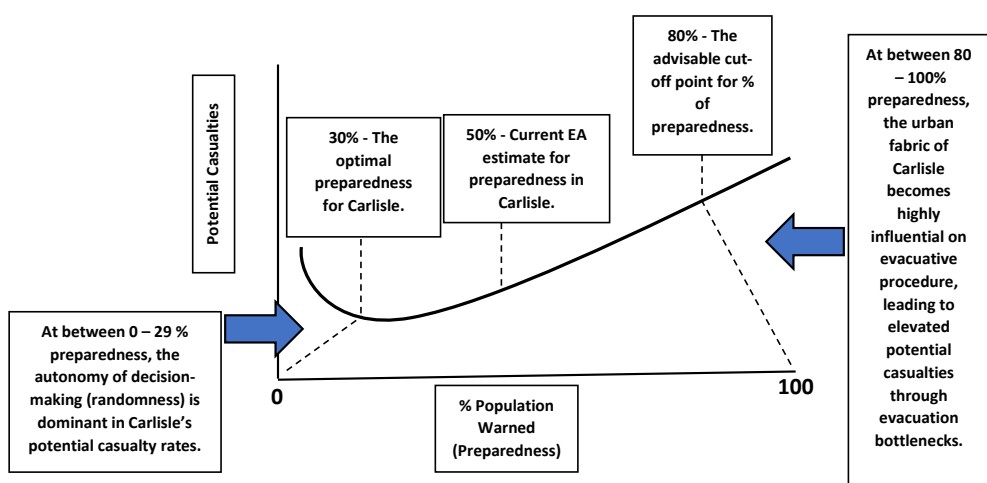

**Figure 11:** A representation of the key results shown in Figure 9 together with concepts that can be associated
with them. It is expected that these percentages will vary with model parameterisation and changes in the area
modelled.

5. **Discussion**

From further interpretation of figures 8 a & b, 9, 10 and 11 it is reasonable to infer that the
agents within the HABM, representing the local population of Carlisle, demonstrate a further
*preference* for basing their response to a flood event on interaction with their surrounding
neighbours, a social response, rather than acting directly from policy instruction. The rate of
innovation (**8a**) impacts less of the Carlisle population over a greater duration compared the
rate of imitation (**8b**). It is believed that this could be because there is a higher number of the
influential, *aware* or *pre-prepared*, agents leaving the vicinity of the flood prior to, or in the
early timesteps of, flood onset and so the message of adoption from these agents becomes
less likely to diffuse through the rest of the population (seen in figures 5 to 7). Conversely,
when the remaining proportion of the population begin to experience the effects of the flood
and a greater number of this population's daily routine becomes disrupted, a greater number
of this population will transition to the choice scenario (figure 4) and begin *checking* with


those agents around them about what an appropriate response will be. This proliferates the
imitative process of evacuation and so would explain why the rate of imitation is more
influential over a shorter period, particularly when the compact social network of Carlisle,
facilitated by a relatively constrained urban topology and morphology; is considered.
A likely explanation of the slightly better correspondence between the curves of 8a compared
to 8b is that they represent a direct instruction at the outset of the simulation and so there is
less time for choice to be considered, with agents taking direct action as soon as possible. The
issue with this is that the agents carrying the innovative knowledge will encounter less agents
as the event unfolds over time, having taken evacuative action from the outset and left the
area where the rest of the agents may not have encountered the flood inundation yet and
are therefore continuing with their daily routine. Consequently, when the function of **(q)** is
considered, a more effective and efficient process for diffusing the evacuative information
amongst the modelled population of agents is seen. To understand why this is the case one
must consider the dynamics at play in a broad sense, **(q)** is a descriptor for internal influence
and, within the HABM, is reliant on agent-agent interaction whilst **(p)** is the innovative
directive from a distal governmental agency which is reliant on engagement from the
population and so to simplify this process as much as possible for these simulations, this
directive was designated as an instruction to 'take evacuative measures immediately'. Worthy
of note here is that, for the applied parameters, the Bass Model is considered a pessimistic
forecasting tool with more optimistic alternatives, which have potential for application in
similar scenarios, being the shifted Gompertz and Weibull distributions, both of which have
superior forecasting and theory testing capabilities but do not offer such a balance between
normative and non-normative interpretation, necessary for this format of analyses, as is the
case with the use of the Bass Model (Jakkola, 1996).
Within the HABM specifically, the format for agent distribution and seeding is more
generalised, and the framework of the daily routine is more direct, than in comparable
models. This is, in some ways, a concession in relative precision, justified by the sustainable
operation of the model within the NetLogo format (Rasmussen et al., 1994; Wilensky & Rand,
2015; Wong & Luo, 2005). Furthermore, with the primary application of this model being
concentrated on the development of understanding regarding the complex nature of human
interaction with the urban and natural environments, under extraordinary or unusual
circumstance, the production of interpretable metrics using a new, interdisciplinary tool is
considered to be a significant first step in enhancing understanding in this area. The general
form of complexity explored in this paper has certainly been subject to greater scholarly
interest in recent times and this has been evident through the proliferation of publications on
the subject and related phenomena, particularly through the last decade (Liu et al., 2015). As
a result of this, complexity science has increasingly undergone a process of extension into
quite different scientific fields (Alexander, 1980; Axelrod, 1970; Wilesnky & Rand, 2015)**.** This
process, whilst a necessary element of scientific progress, has in some way acted to separate
theory from application and has led to a diminished emphasis on cross-disciplinary
applicability, leaving potentially useful scientific tools isolated or limited by the technological
capability of the time. This has furthered the highly fragmented development of agent-based



models and modelling frameworks (Axelrod, 1970; Müller et al., 2013; Namatame, 2016).
These largely fall into one of two polar groups: those which over-emphasise a very specific
use through a reductive process of refinement to meet validative expectations, or those which
place themselves at the extremity of validation because of the physically unimaginable
complexity that is being modelled. It is here, despite any shotcomings, where the truth of the
HABM is found; at the point of bifurcation between these groups  (Assaf & Hartford, 2002;
Eberlen & Scholz,2017; Guo et al., 2008; Liu et al., 2015; Morss et al., 2016; Nunes Correia et
al., 1998; Waldorp et al., 1993; Wei et al., 2003; Werrity et al., 2007).

The provision of a probabilistic framework (**figure 4**) for the 'pseudo-random', this being the
large array of numbers underlying the agent's movements within the model environment,
which are in effect limitless but are bounded by fractal (self-replicating) 'stochasticity' of the
model layer implemented within NetLogo, agents to interact with (i.e. leave, stay, etc.), has
great importance for the general and trans-disciplinary application of the methods in this
paper. This is particularly the case in the absence of empirical certainty for how the real
population of Carlisle might individually act on the day. But the framework provides some
necessary, general, parameters for human response in the event of a flood and so greatly
reduces the possibility of an entirely chaotic modelling scenario, whilst also maintaining a
realistic representation of choices that represent systematic functions of the community,
infrastructure, and environmental dimensions within the urban and flood hazard system.
Finally, it allows reproducibility for the HABM where components of future hydro-sociological
models could simply be substituted for those of the HABM (Landstrom et al., 2011, Sabatier,
1986; Wong et al., 2005).

In reality, the social elements of the complexity explored here are as unpredictable as they
are dynamic: this challenges forecasting behaviours in addition to understanding. As
evidenced in this paper, the social elements are represented by many different participants
who adapt and influence one another, interacting in intricate ways that continually reshape
their individual and collective responses. When performed collectively, these interactions
form systems which are characterised by multi-scale interactions between the micro
(individual) to the macro (demographic, economic and governmental). The collective
coalescence of multi-scale interactions have been termed 'Complex Adaptive Systems' and
they have a significant underpinning from research focused on their inter-disciplinary and
methodological design so as to better understand the significant challenges presented by
their complexity (Dugdale et al., 2009; Gilligan et al., 2015; Holland, 2014; Liu at al., 2015;
Morss et al., 2016.)

Ultimately, the design of, "holistic risk management strategies requires an accurate
understanding of the level of risk across the various layers of society. One important
remaining limitation in our understanding of flood risk is the way individuals perceive and
respond to risk. Even if we manage to model population density and flood inundation with
increasing accuracy, assumptions about peoples' risk reducing behavior, willingness to
relocate, and access to information play a key role in the actual level of risk" (Jongman, 2018).
Individual perception is an extremely complex phenomena and representing this from event



and systematic complexity is paramount for developing further understanding of the nature
of the physical-social interactions discussed here, so that evacuations may be better
organised and the greatest number of lives may be saved in the event of a complex event, like
a flood (Berendracht et al., 2017). Consequently, the non-linear characteristics associated
with complex adaptive systems, including influential systematic processes such as
heterogeneity, phase transition and emergence, require that our methods, such as those
illustrated in the HABM, also attempt to represent the general complexity of adaptive
systems. Given that such systems exist as macro networks of partially connected micro
structures (fundamentally via individuals interacting in different groups which adapt to
changes in the surrounding environment), the methods must then also include microscale
models which are able to simultaneously simulate cross-scale operations, interactions and
responses amongst multiple participants (Assaf & Hartford, 2002; Dawson et al., 2011), to
provide interested parties with access to more representative insights of what is and could
be unfolding in reality.
Finally, during the 2005 flood, as modelled by the HABM for this paper, three deaths occurred.
During the 2015 flood event in Carlisle, the River Eden exceeded the 2005 flood level by
600mm, yielding only one fatality but with a much greater economic impact (The Environment
Agency, 2016). Even with the  generalised 'potential fatality' metric implemented into the
HABM, set as such due the low number of actual fatalities which occurred during the 2005
event, if the results of the model's simulations are to be believed; then there is a much greater
potential for a fatal impact within the flood inundation area than that which presented itself
during the actual events of Carlisle in 2005 and 2015. Here, the true importance of the HABM
and Bass Model results is that they offer a counter-intuitive scenario to be further
deliberated, one which could prove significant for flood hazard management in Carlisle and
risk management overall.
6.  **Conclusion and future development**
This        paper       began       by        proposing      two        specific      questions:
**1. During a flood, does site-specific urban topography and morphology, change the**
**optimum evacuation warning strategy?**
**2. Do people (agents) respond better to direct or indirect (word of mouth) evacuation**
**warnings during a flood event?**
These questions were formulated to explore the UK governmental shift towards alternative,
bottom-up, action for addressing flood vulnerability and risk, as especially affected by agent
response and urban morphology. These objectives simplify what is a very complex scenario
and so with respect to this complexity, a methodological framework for addressing these two
objectives was formulated and demonstrated, producing results via a coupled hydrodynamic
and agent-based model: the HABM. This model was used to explore the complexity of human



responses and behaviours during a flood event with a view to better specifying the two basic elements of the flood hazard system, a physical flood interacting with a human urban system. From this investigation, a range of implications were uncovered by the model simulations of response and behaviour. Based upon observation of these implications, some practical recommendations can be made for flood warning delivery and strategy as follows:

- Agents operating within a system of change show a preference for action via a socially *imitative* process as opposed to one which operates from *innovation*. This would suggest that bottom-up approaches towards warning and evacuation would benefit from incorporating measures that harness this understanding of group processes.
- Owing to the influence of site topography on the outcomes of social response, and the creation of potential congestion points within affected sites, a phased response to flood events should be an actionable option within flood warning strategy and delivery.
- During the process of issuing a flood event warning, the geography (topography and morphology) of the affected site can significantly influence the success or failure of the evacuative process and so due attention to this influence should be given during planning. This reaction phase involving the response and movement of people does not normally receive much attention and likely should.
- Whilst it might be a desirable goal to achieve a 100% preparedness within a flood-prone area, the results from the HABM simulations suggest that this may not be necessary, or even desirable. Simulations support the idea that the 50% estimate of the EA for Carlisle is the best value for efficient evacuation, owing to the social dynamics and the topography of the site. The design of 'optimal' impacts for a ranging of percentages of prepared people, and for sites with differing layout and population dynamics, needs to be critically considered in future flood response strategies.

There are significant questions that arise from these recommendations which require further analysis. Enhanced development of the HABM and the related themes will look to provide this further analysis in the form of the following:

- The nature of the agent decision-making process in locations where interaction is concentrated, e.g. is social response hastened where there is a higher population density?
- The nature of agent response with respect to the physical attributes of the flood event, e.g. attenuation of the flood hydrograph & variations in flood volume influencing the process of evacuation.
- Different urban morphologies: will these give dramatically different results to those produced for Carlisle?

Whilst not a predictive tool, the implications of the results herein outlined, coupled with such future developments of the HABM, are useful in providing greater scope for including and quantifying relevant operative factors that are involved in flood vulnerability, risk and


resilience as related to urban systems. The HABM offers a dynamic method for simulating
important actions linked to these, with the potential to enhance quantitative analyses in
support of the decision-making process for flood hazard management. This paper
demonstrates that such quantification can involve not only flooding itself, but also potential
human responses. These may exacerbate the risk if they are not accounted for during
planning, or they may be diminished through improved response planning. Other hazard
environments may similarly be analysed using the approach here outlined, providing many
points of further discussion and consideration for stakeholders involved with risk assessment.
The HABM can be a useful analytical tool for supporting and expanding on these points
moving forward.

**Data availability:** The population data was accrued and modified from the 2011 aggregate
NOMIS (ONS) database found at: https://www.nomisweb.co.uk/census/2011
This was cross-referenced with the supporting flow data found at:
https://wicid.ukdataservice.ac.uk/
Building footprint data was taken from OSM, copyrighted to OpenStreetMap contributors
and available from: https://www.openstreetmap.org/
The LISFLOOD dataset for Carlisle can be requested directly from Dr. Jeffrey Neal with
further details on the LISFLOOD-FP available at:
http://www.bristol.ac.uk/geography/research/hydrology/models/lisflood/
Bass Model curves were informed by information found on The Bass's Basement Research
Institute webpage, © 2008, 2009, 2010 Bass's Basement Research Institute, at:
http://www.bassbasement.org/BassModel/Default.aspx
The prototype Netlogo code for this model is currently still being used and modified as an
active component of Thomas O'Shea's PhD thesis but it will be made available via open-
source repository on the NetLogo Modelling Commons page at:
http://modelingcommons.org/account/login under the title of this paper.

**Author contributions:** Thomas O'Shea wrote this paper with assistance and input from Paul
Bates and Jeffrey Neal.

**Competing interests:** The authors declare that they have no conflict of interest.

**Acknowledgements:** The authors are indebted to Professor John Lewin, Toon Haer and his
colleagues at the IVM, VU Amsterdam, Professor Nobuhito Mori and his colleagues at the
Research Division of Atmospheric and Hydrospheric disasters, DPRI, Kyoto and our Bristol
colleagues, Laurence Hawker and Jeison Sosa Moreno for their helpful thoughts on the pre-
developmental stages of the HABM network.

**Financial Support:** Thomas O'Shea is supported by the EWS Educational Trust Exceptional
Contribution Award. Paul Bates is supported by a Royal Society Wolfson Research Merit
Award and Jeffrey Neal is supported by a NERC fellowship for interdisciplinary research on
flooding in Vietnam.



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
