# Peer review of "Thomas O'Shea1, Paul Bates1 and Jeffrey Neal1"

_Natural Hazards and Earth System Sciences, 2019_

## Referee Comment (RC1) · Anonymous Referee #1 · 23 Dec 2019

General comments

This paper recognises the complexity of hazard situations and responses, but also that adaptive actions overall may be simulated from individual or 'agent' behaviours through using agent-based models (ABMs). On the physical side, hydrodynamic behaviour can have an equivalent concern for the local through detailed topographic modelling and floodwater routing. The paper demonstrates how combining the two, through an innovatively developed approach coupling hydrodynamic and agent-based models (named here HABMs), allows site-specific procedures for warning provision and evacuation to be usefully designed. This is accomplished through simulating populations and exploring their alternative behaviours to see which might be of most benefit for responses to flooding events given the local geography – as in the case of Lancaster, UK, flooding described here. An interesting feature of the approach is that the human behavioural aspects are here justified by appeal to social theory, just as the (now better-established)

[Figure]

hydrodynamic modelling is justified and rests on physical theory.

Specific Comments

The promotion of new quantitative approaches that combine physical understanding of hazards with possible actualities of human responses to them is surely to be welcome. Until recently there has commonly been an academic gap between the two: (1) improved modelling of physical phenomena and their dynamics on the one hand, but (2) 'top-down' imposition of (mostly hard engineering) solutions at affected sites without exploring what their populations might be doing, or could best be doing, in response. Localized decision-making is likely to improve greatly if those involved have good understanding of what best to do in the situation they confront – rather than putting schemes to the vote at some higher political level, the advantages or disadvantages of which are little understood on the ground. 'Participatory methods' have to be better than this. Coping with hazards is at heart a human cognitive activity, and so how people at different participatory levels can behave, or get informed as to how better to behave, should be beneficial.

Technical Corrections

18. ' . . constructed using the Bass Diffusion Model'. 118. Omit last comma in cited references. 127. Put 'Dawson' in the bracketed reference. [also line 570] 171. Dawson et al., 2011; Müller, . . . 183. Semi-colons needed between EA Reports: 2006; 2012; 2016. And after Neal et al., 2009; [Also in line 752 after 2003.] No need for 'ands' within the reference brackets; see also lines 207-8; 214; 234; 267. 304. 'imitators' 305. a priori. 323. Chen & Zhan, 2008; 334. One quote mark only needed. 394. Dawson et al., 2011; Also no full stop after 1994 within the brackets. 487. Comma after information [to be consistent with elsewhere]. 587. Add full stop to sentence after bracket. 627. How does the map show 'areas through which people are most likely to move' as the caption suggests? That's made more visible in figure 10. 651. Why a semi-colon here? Perhaps: '. . traditional terms that may be thought of as an acceptance' 654. Give cited

reference, not just its number. [Also line 665] 659. Semi-colon needed after 2015. 719. Last sentence of caption incomplete. 751. Omit 'of'. 758. Full stop after bracket, not before it. 846/7. (Figure 8a), (Figure 8b) [add the word 'figure', and not in bold]. Also line 859. 875. 'being based on shifted Gompertz . . ' 902. 'value' rather than 'truth' perhaps. 907. Not bold. Need to check house style (especially whether 'figure' should have a capital letter). This long sentence at the start of the paragraph needs recasting, too. 923. their understanding. 940. Readers might appreciate a page number for this quotation.

Lettering sizes on Figures 2, 3 and especially the side panels of Figures 5-7 are on the small size.

---

## Referee Comment (RC2) · Anonymous Referee #2 · 2 Jan 2020

This paper proposed an innovative approach to represent the complex human behavior during flood evacuation in Carlisle by combining a hydraulic model (LISFLOOD-FP) and an Agent-Based Model (NetLogo). I have really liked the idea of using the Bass Diffusion Model to represent the agent's behavior during flooding. The results of this study demonstrated the importance of using a holistic approach to flood management purposes. Overall, I have enjoyed reading the paper and I found the manuscript well-written, clear, and results are properly described and discussed. For this reason, I do recommend a minor revision before this paper can be accepted in NHESS. However, I still have a few comments which I hope will be useful to the author to strengthen the manuscript.

1) It looks to me that one important aspect of the ABM is not included in your approach, i.e. the traffic model. In fact, during the evacuation process, traffic congestions can play a crucial role before the agents select to respond to the flood all at the same time. How-

ever, it is not clear to me what are the dynamic characteristics of the agents 'movement (e.g. speed) and how are the road features included in the ABM. In fact, evacuation strategies may change based on the direction, capacity, and maximum allowed speed of the road network in Carlisle.

2) Why did the authors coupled Lisflood with the ABM if societal actions will not influence flood propagation (at least in this study)? Of course, the proposed coupling framework can allow simulating more complex situations, e.g. placing sandbags or other tools to protect from flooding, but it will drastically increase computational costs. I assume that such costs may reduce if the raster files are uploaded within the NetLogo framework each simulation time step. Moreover, what is the computational time for 1 simulation?

3) How the working locations for all the agents are assigned? From what I could understand from the manuscript, the daily routine is randomly assigned at each simulation based on the census information of the specific commercial area in Carlisle for 2005. Is this valid also for the working locations?

4) When an agent receives the warning and decides to act immediately it will then exit the DEM using available network road. Is this a realistic situation? If yes, please provide a reference to support your choice.

5) Can you provide an example of the "innovative knowledge to respond to the flood upon onset" that a pre-prepared agent can use? (line 579) Maybe I have missed some details

6) Besides for the DEFRA estimation for Carlisle at line 756-758, did you evaluate the model results with other observation data (e.g. tweets or report for some specific parts of the city)? I have found some (maybe useful) information in this webpage http://www.intrescue.info/hub/index.php/carlisle-floods-8th-january-2005/

7) The authors stated that "The only study to date to drive an ABM with a hydrodynamic

model was that of Dawson (et al., 2011)." This is not totally correct. Also in Medina et al. (2016) an ABM and a hydraulic model were coupled to test large scale evacuation strategies in coastal cities under threat of imminent flooding due to extreme hydro-meteorological events. Moreover, other studies coupled ABM with a hydraulic model for flood risk management purposes (Abebe et al., 2019).

8) Try to improve the quality of figures 8 and 9

Reference:

- Abebe Y.A., Ghorbani, A., Nikolic, I., Vojinovic, Z. and Sanchez, A. (2019) A coupled flood-agent-institution modelling (CLAIM) framework for urban flood risk management, Environmental Modelling & Software, 111, 483-492.

- Medina, N., Sanchez, A. and Vojinovic, Z. (2016) The Potential of Agent Based Models for Testing City Evacuation Strategies Under a Flood Event, Procedia Engineering, 154, 765-772, https://doi.org/10.1016/j.proeng.2016.07.581.

---

## Referee Comment (RC3) · Anonymous Referee #3 · 5 Jan 2020

This paper attempts to present an integrated hydraulic-ABM model for modelling individual behaviour during flooding. Human interventions could significantly affect flood risk even during an event, especially in densely populated urban areas. This research represents an encouraging attempt to develop an approach to model human activities in the city of Carlisle during a flood event in 2005, which is an innovative and necessary step forward in flood risk assessment. But at its current form, the paper is difficult to follow, and it is not clear what the core focus and innovation is. It must be substantially revised and improved before accepting for publication. Hope the following comments will help the authors revise their paper.

The major concerns: 1. What is the major novelty or focus of this work? Is it the 'new' modelling framework? Or is it the application of the model to understand human

activities during a flood event in the case study?

"This paper presents a new flood risk behaviour model developed using a coupled Hydrodynamic Agent-Based Model (HABM)", which suggests the modelling framework is the key novelty in this work. But the presented HABM takes offline modelling outputs (flood depth) from LISFLOOD-FP to drive the agent-based model developed in the NetLogo framework. This is actually a 'step backwards' from the modelling approach as reported by Dawson et al. (2011), in which "a hydrodynamic model simulates the floodwave was also developed within the ABM platform and interacts directly with the agents and the built environment". The authors argued that "this study initially coded the hydrodynamic model directly within the ABM meaning advantage could not be taken of recent developments in efficient numerical methods for solving the shallow water equations . . . and high-performance computing . . ." But since the current modelling framework does not actually 'couple' with the 'more advanced' LISFLOOD-FP with the agent-based model, the modelling framework itself does not present any novelty in terms of numerical development or method. The approach of using offline flood modelling outputs to drive an agent-based model has also reported in the literature, e.g. Lumbroso et al. (2011) developed a life safety model to estimate risk to people imposed by dam breaks or flash floods. In their work, their Life Safety Model could use outputs from any available two-dimensional hydrodynamic models that solves the shallow water equations (e.g. Telemac-2D, TuFlow) or the simplified forms (e.g. LISFLOOD-FP).

If the focus of the paper lies in the application of the model to understand food-driven human dynamics in the case study. There is no strong evidence showing the model settings reflect reality and so the results and the conclusions may be misleading.

2. Following the above comments, it is difficult to be convinced that the model settings can represent actual human dynamics during a flood event in Carlisle since 1) the behaviour rules for individual are over-idealised and there is no evidence to back the choice of behaviour rules; 2) the communication rules between agents are also over-simplified, e.g. how are text, social media and other forms of wireless communications

taken into account, which may significantly affect the simulation results; 3) traffic systems and key organisations are not represented in the model which will inevitably have significant influence on the results and conclusions; and 4) the model results were not validated at all. Therefore, the results and the conclusions from the simulation may not be valid and may be actually misleading.

Minor issues: 1. Why the authors use the 2005 flood event but not look at the more recent 2015 event? More information would be available from different sources for the more recent event to inform and validate human activities. 2. The paper is difficult to follow and the authors should more explicitly explain the modelling framework, how the agents are interact and communicate, and how the behaviour rules are set and why, etc. 3. Since the human activities do not have any impact on the flood dynamics and the agent-based model is only driven by offline flood model outputs, it is NOT a 'coupled' model. 4. The title, 'an agent-based model for flood risk warning', is a bit confusing. Based on its current capacity, the model cannot be used for 'flood risk warning'.

References: Dawson, R., Peppe, R. & Wang, M., 2011, An agent based model for risk-based incident management of Natural Hazards. Nat. Haz., 59(1): 167-189. Lumbroso, D.M., Sakamoto, D., Johnstone, W.M., Tagg, A.F. and Lence, B.J., 2011. Development of a life safety model to estimate the risk posed to people by dam failures and floods. Dams and Reservoirs, 21(1): 31-43.

---

## Author Comment (AC1) · 13 Jan 2020

RC1 – Response

The authors would like to extend their sincere thanks to the referee for their time and considered thoughts on the submission. All comments and corrections have been thoroughly considered, with our respective action and/or response to these outlined below.

General comments:

This paper recognises the complexity of hazard situations and responses, but also that adaptive actions overall may be simulated from individual or 'agent' behaviours through using agent-based models (ABMs). On the physical side, hydrodynamic behaviour can have an equivalent concern for the local through detailed topographic modelling and

floodwater routing. The paper demonstrates how combining the two, through an innovatively developed approach coupling hydrodynamic and agent-based models (named here HABMs), allows site-specific procedures for warning provision and evacuation to be usefully designed. This is accomplished through simulating populations and exploring their alternative behaviours to see which might be of most benefit for responses to flooding events given the local geography – as in the case of Lancaster, UK, flooding described here. An interesting feature of the approach is that the human behavioural aspects are here justified by appeal to social theory, just as the (now better-established) hydrodynamic modelling is justified and rests on physical theory.

Author response: The authors appreciate the referee's acknowledgement of their attempt to innovate an approach towards the development of useful designs for warning provision and evacuation that harness the detailed elements of physical and social theory together. The authors agree that physical theory is, currently, the better-established format for justifying action with respect to warning provision and evacuation but also believe that there are influential degrees and actualities of the warning and evacuation processes which are not yet, or cannot be, accounted for within physical theory. It is here where the authors hope their attempts to illustrate the potential influence of these unaccounted factors, through the lens of urban flooding and the HABM framework, find the greatest value.

Specific Comments:

The promotion of new quantitative approaches that combine physical understanding of hazards with possible actualities of human responses to them is surely to be welcome. Until recently there has commonly been an academic gap between the two: (1) improved modelling of physical phenomena and their dynamics on the one hand, but (2) 'top-down' imposition of (mostly hard engineering) solutions at affected sites without exploring what their populations might be doing, or could best be doing, in response. Localized decision-making is likely to improve greatly if those involved have good understanding of what best to do in the situation they confront – rather than

putting schemes to the vote at some higher political level, the advantages or disadvantages of which are little understood on the ground. 'Participatory methods' have to be better than this. Coping with hazards is at heart a human cognitive activity, and so how people at different participatory levels can behave, or get informed as to how better to behave, should be beneficial.

Author response: The authors roundly agree with this assessment and hope the essence of this agreement can be felt from reading the submitted paper. De Groot and Schuitema (2012) suggest, quite robustly, that there is a distinct link between the acceptability of environmental policies, social norms and the characteristics of those policies, further to which, Kinzig (et al., 2013) suggests that the insufficient insight on the coevolution of these norms and policy instruments is what compromises the ability of decisionmakers to craft effective solutions to society's most intractable environmental problems. The authors recognise the growing annual losses attributable to the environmental problem of flooding as an extension of this lack of insight and as having a solution in the analysis, evaluation and development of participatory methods which are equally informed by both physical and social theory. This paper and example therein serve as a vehicle for this sentiment and the authors hope the approach outlined in the paper serve as a catalyst for the development of further hybrid narratives that are necessary for the advancement of effective participatory methods and policy.

Technical Corrections:

Author response: The referee's direction for technical corrections throughout the submission are very much appreciated by the authors and these have been implemented within an updated version of the manuscript to be uploaded following the period of interactive discussion.

18. '... constructed using the Bass Diffusion Model'. 118. Omit last comma in cited references. 127. Put 'Dawson' in the bracketed reference. [also line 570] 171. Dawson et al., 2011; Müller, . . . 183. Semi-colons needed between EA Reports: 2006; 2012;

2016. And after Neal et al., 2009; [Also in line 752 after 2003.] No need for 'ands' within the reference brackets; see also lines 207-8; 214; 234; 267. 304. 'imitators' 305. a priori. 323. Chen & Zhan, 2008; 334. One quote mark only needed. 394. Dawson et al., 2011; also, no full stop after 1994 within the brackets. 487. Comma after information [to be consistent with elsewhere]. 587. Add full stop to sentence after bracket. 627. How does the map show 'areas through which people are most likely to move' as the caption suggests? That's made more visible in figure 10. 651. Why a semi-colon here? Perhaps: '. . .traditional terms that may be thought of as an acceptance' 654. Give cited reference, not just its number. [Also line 665] 659. Semi-colon needed after 2015. 719. Last sentence of caption incomplete. 751. Omit 'of'. 758. Full stop after bracket, not before it. 846/7. (Figure 8a), (Figure 8b) [add the word 'figure', and not in bold]. Also line 859. 875. 'being based on shifted Gompertz. . .' 902. 'value' rather than 'truth' perhaps. 907. Not bold. Need to check house style (especially whether 'figure' should have a capital letter). This long sentence at the start of the paragraph needs recasting, too. 923. their understanding. 940. Readers might appreciate a page number for this quotation. Lettering sizes on Figures 2, 3 and especially the side panels of Figures 5-7 are on the small size. _______________________________________________________________

References:

De Groot, J.I.M. & Schuitema, G., How to make the unpopular popular? Policy characteristics, social norms and the acceptability of environmental policies. Environmental Science & Policy, 19-20, 100-107, DOI: https://doi.org/10.1016/j.envsci.2012.03.004, 2012. Kinzig, A.P., Ehrlich, P.R., Alston, L.J., Arrow, K., Barrett, S., Buchman, T.G., Daily, G.C., Levin, S., Oppenheimer, M., Ostrom, E. & Saari, D., Social Norms and Global Environmental Challenges: The Complex Interaction of Behaviours, Values, and Policy. BioSciences, 63-3, 164-175, DOI: https://doi.org/10.1525/bio.2013.63.3.5, 2013.

―――――――――――――――――

---

## Author Response (AR1)

**An agent-based model for flood risk warning.**

**Thomas O'Shea[1], Paul Bates[1] and Jeffrey Neal[1]**

**[1]** School of Geographical Sciences, University of Bristol, UK.

**Correspondence**: Thomas O'Shea (t.oshea@bristol.ac.uk)

**Abstract**

This paper presents a new flood risk behaviour model developed using a coupled Hydrodynamic Agent-Based Model (HABM). This model uses the LISFLOOD-FP Hydrodynamic Model and the NetLogo (NL) agent-based framework and is applied to the 2005 flood event in Carlisle, UK.  The hydrodynamic model provides a realistic simulation of detailed flood dynamics through the event whilst the agent-based model component enables simulation and analysis of the complex, in-event social response. NetLogo enables alternative probabilistic daily routine and agent choice scenarios for the individuals of Carlisle to be simulated in a coupled fashion with the flood inundation. Experiments are also constructed using a novel, 'enhanced social modelling component', comprising the Bass Diffusion Model, to investigate the effect of direct or indirect warnings in flood incident response.

From the analysis of these coupled simulations, management stress points, predictable or otherwise, can be presented to those responsible for hazard management and post-event recovery. The results within this paper suggest that these stress points can be present, or amplified, by a lack of preparedness or a lack of phased evacuation measures. Furthermore, the methods here outlined have the potential for application elsewhere to reduce the complexity and improve the effectiveness of flood incident management. The paper demonstrates the influence that emergent properties have on systematic vulnerability and risk from natural hazards in coupled socio-environmental systems.

> **Commented [TO1]:** Technical correction Ref #1 18. '... constructed using the Bass Diffusion Model'.

**1. Introduction**

Flood hazard, or flood incident, management is a challenge that incorporates aspects of the natural sciences (hydrology, ecology, etc.), the social sciences (economics, politics, psychology, culture, etc.) and engineering. It is important for the efficiency and efficacy of decision-making processes to recognise that decision-making during floods involves what has been termed "technical complexity" (Nunes Correia, Fordham, Da Graca Raravia & Bernardo 1998). Specifically, this is the social response to the hazard, and encompasses interactions between individuals, the diffusion of decision-making and collective, during-event, behaviours (Larsen, 2005). This complexity cannot, either theoretically or physically, be eliminated when planning for flooding incidents (Assaf & Hartford, 2002; Bennet & Tang, 2017; Correia, Rego, Saravia & Ramos, 1998 and Dawson, Peppe & Wang, 2011) and can be a threat to effective planning processes (Axelrod, 1970; Nunes Correia et al., 1998).  In a broader sense, this complexity is a measure of the scale of the interactions within the affected area, encompassing dynamic multi-scale interactions and adaptions between individuals, groups, infrastructures, government and the economy, all contributing to the social, political and physical aspects of flood hazard management (Dugdale, Saoud, Pavard, & Pallamin, 2009; Fordham, 1992; IPCC, 2014; Kossiakoff & Sweet, 2002; Werrity, Houston, Ball, Tavendale & Black 2007 and Wisner, Blaikie, Cannon & Davies, 1994).

Recent decades have seen strong emphasis being placed on multi-scale, *participatory* methods for dealing with floods resulting in a paradigm shift from *flood defence* to *flood risk management* (Assaf & Hartford, 2002, Dawson et al., 2011, DEFRA, 2007; IPCC, 2014 and Wisner et al., 1994). Such participation means the inclusive involvement of individuals and multiple agencies in the processes of hazard management, policy implementation and post-event recovery. This emphasis is logical in that it aims to incorporate, as far as possible, the requirements of all those involved in the hazard planning process across a scale hierarchy that passes from government bodies to emergency services, and on to the affected individuals themselves. The complexity of such an ideal becomes apparent given that the intricate natures of human environments and environmental dynamics are, to a large degree, perceived as independent, and that when the two come into contact, complexity becomes amplified within a coupled socio-environmental system. For example, between 2010 and 2015, UK Government policy for flooding underwent a transformation that sought to address some of the known complexities of flood incident management (DEFRA, 2007; Eberlen, Scholz & Gagliolo 2017; The Environment Agency, 2012 & 2016). The UK Government's Department for Environment, Food & Rural Affairs (DEFRA) national framework for flood management emphasises the importance of localised decisions about flood risk and makes suggestions for developing community-based solutions to manage flood risk on a finer spatial scale. This transformation emphasised the need for innovative new approaches to managing the localised risk of flooding. This was expected to provide the foundation for better management at the larger scale as 'good practice' innovations spread across more communities. Thus, UK flood policy can be defined as moving from a top-down to bottom-up approach, often referred to as '*alternative action*' (DEFRA, 2007; Kossiakoff & Sweet, 2002).

**Commented [TO2]:** 1 & 2 i) Narrative enhancement Ref #3
Included reference for relating diffusion concepts to the modelled process.

[revised manuscript text omitted]

**Commented [TO8]:** Technical correction Ref #1 304. 'imitators'

**Commented [TO9]:** Technical correction Ref #1 305. a priori.

**Commented [TO10]:** Technical correction Ref #1 323. Chen & Zhan, 2008;

Model assumes that $M$ is constant, though in practice and over longer periods, M is often
slowly changing according to population change and product memory.
• $(p)$ – The coefficient of innovation, so-called because its contribution to new adoptions
does not depend on the number of prior adoptions. Since these adoptions are due to
some influence outside the social system, the parameter is also called the 'parameter of
external influence.'

**Commented [TO11]:** Technical correction Ref #1 334. One quote mark only needed.

• $(q)$ – The coefficient of imitation has an effect that is proportional to cumulative adoptions
A(t), implying that the number of adoptions at time t is proportional to the number of
prior adopters. In other words, the more that people talk about a product, the more other
people in the social system will adopt it. This parameter is also referred to as the
'parameter of internal influence'.
The other variables in the Bass Model relationship and calculated from $M$, $p$, $q$ and $t$, are:
• *f(t)* - The portion of M that adopts at time t,
• *F(t)* - The portion of M that have adopted by time t,
• *a(t)* - The adopters (or adoptions) at t,
• *A(t)* - The cumulative adopters (or adoptions) at t.
The outcomes of the coupled application of these three components (sections 2.1, 2.2 & 2.3)
towards the two objectives are further illustrated in section **4** and are discussed further in
section **5**.
Of further interest here is how to qualify the communication taking place within the HABM.
In sociological terms, the imitative process involved is broadly one of inter-agent
communication and collective response. According to the sociologist Gabriel Tarde and his
Laws of Imitation (Tarde, 1903), as applied to 'groups of people', innovations must undergo a
process of diffusion over time to gain a foothold and become a component in the decision-
making process linked to the innovation, be this *adoption* or *rejection*. Tarde's process
involved in the diffusion of innovation has undergone some revisions in the decades since
being first proposed and can now be defined through the following five steps:
• First Knowledge,
• Attitude formation,
• Adoption or rejection,
• Implementation,
• Confirmation of the decision.
Via the Bass Model, the HABM for Carlisle allows a simulated engagement with the first four
steps of Tarde's process, the fifth being confirmed in the representation of the first four
activities as the simulation advances over time. This interpretation of social imitation and
adoption was used as a basis for investigating the influence of these processes in an event
where time is relatively constrained and the stakes of action are high, such as during a flood onset. The values for this process of adoption were taken from the change in overall un-
prepared population in Carlisle transitioning to a 'prepared state' based upon contact with a
'pre-prepared', or innovative, agent. This transition was represented by the percentage of the
population in possession of the appropriate knowledge for effective flood evacuation who
then reported this change back as an agent-orientated change of state throughout the
simulation of the flood. This rate of change of state is then fed into the Bass Model functions
to produce diffusion curves like those seen in figures 8a & b and discussed in further detail in
sections 4 and 5.

3. **Core model construction and system dynamics**

Given the complexity caused by the incorporation of these diverse elements within
considerations of a flood hazard system, the benefits of a standardised flood incident
management strategy based on an understanding of these dynamics might not be
immediately apparent. Further management of complexity might necessarily arise through
the required interactions between the individuals and organisations who might very well have
conflicting interests linked to contrasting elements in their expertise or experience (Hart,
Nilsson & Raphael, 1968; Hornor, 1998). Furthermore, the feedbacks within a flood hazard
system, particularly an urban one, can lead to a spectrum of dampening and amplification of
behaviours within the system, the dynamics of which could be influential on outcome, yet
difficult to account for in a standardised flood incident management strategy (Assaf &
Hartford, 2002; Dawson et al., 2011; Rasmussen, Pejtersen and Goodstein, 1994.) It is here
where the HABM concept reaches out to the concepts of phenomenology, poststructuralism,
structuration theory, structural functionalism and symbolic interactionism to inform the
conception of a modelling framework that incorporates the important social notions of these
disciplines and thus anchors the modelling element of the HABM to the cardinal philosophical
and sociological concepts underlying it and the outputs produced. The appeal of this approach
lies primarily in the novelty of the undertaking in addition to the application of concepts from
disciplines such as sociology, philosophy and psychology, which complement the model by
offering access to new terminology and theoretical bases for better representing social
systems, focussed on *relatedness* rather than *boundedness* between the dimensions and the
whole (Alexander, 1980) ; within a coupled modelling framework. Here, the benefit of a more
holistic representation can lead to the development of a more effective and holistic
understanding of how to manage social dynamics, responses and functions within physical
models where they can have further impact on effective planning for and outcomes from the
whole system and the components comprising that system (Smith & Tobin, 1979; Zarboutis
& Marmaras, 2005).

With these details in mind, and urban systems being the primary interest in this paper (figure
2), the first step beyond bringing together the initial HABM components was to devise a
conceptual format that describes the key dimensions of the urban system within a
parameterised and reproducible framework. In this paper they will be primarily referred to as
*dimensions*, alternatively they can be called *'sets'* (or *centres* (Alexander, 1980), and can be

**Commented [TO12]:** Technical correction Ref #1 394. Dawson et al., 2011; also, no full stop after 1994 within the brackets.

[revised manuscript text omitted]

**Commented [TO21]:** Technical correction Ref #1 654. Give cited reference, not just its number. [Also line 665]

**Commented [TO22]:** Technical correction Ref #1 659. Semi-colon needed after 2015.

**Commented [TO23]:** Technical correction Ref #1 654. Give cited reference, not just its number. [Also line 665]

but not directly from the external directive (e-mail, text alerts etc.), according to
communication between agents, within the total flood affected population of Carlisle, is more
influential over a shorter duration than the operation of (**p**). The variance between the three
lines would suggest that there is some disagreement between the baseline functions of the
Bass Model differential equation and those for discrete and continuous time for (**q**) and it is
believed that this is likely related to the unusually high value attributed to the 30% likelihood
of agents *agreeing* to imitate the innovative agents and become imitators, as well as the
general stochasticity related to the reliance on 'proximal contact' for communication
between agents, which is likely but not guaranteed in any situation; particularly in one as
frenetic as that involving a flood.

[Figure]

**Y-axis**: Population of Carlisle, **X-axis**: Time (hours.)

**Figures 8 a & b:** Example Bass diffusion curves for p or innovation (top), and q, or imitation (bottom), at Carlisle. Illustrated are the curves for the continuous time Bass Model functions (blue/ M*SM_f & red/ M*f) for discrete and incremental time-steps and the Bass Model differential equation (green/ M*DE-f). The Y-axis for both curves represents the maximum number of individual agents with potential to respond in accordance with the type of warning given and action taken,

**Commented [TO24]:** Technical correction Ref #1 719. Last sentence of caption incomplete.

This bridge between sociological and theoretical concepts of process diffusion, or between
internal and external components, provides insight into the relationship existent between policy and responsive behaviour. Furthermore, the Bass Model's use in the analysis of flood response dynamics is a broadly useful one, providing quantitative evidence of behaviour, in the form of diffusion curves (figures 8a & b) and, for the dynamics of during-event agent communication, thus implementing Tarde's sociological laws into the modelling process. In addition, it represents both the 'innovative' *i.e.* individual response to policy direction, and the 'imitative' processes related to this direction, which certainly have influence on the micro, and potentially macro, scale human responses to flood events (Bernardini, 2017, Guyot & Honiden, 2006).

[revised manuscript text omitted]

**Figure 10:** An aerial image of Carlisle illustrating the preferential direction for escape to the south west along
the A595. Further illustrated are the most prominent chokepoints (**red crosses**) for reduced evacuative flow of
people between 80 and 100% preparedness. These points were identified from the HABM as the nodes in the
street network overlay which have the most consistently high densities of agents throughout the range of
simulations. (Contains OS data © Crown copyright and database right (2019))
As is illustrated in figure 11, with less than 30% preparedness, agents within the HABM  show
a preference for evacuation away from Carlisle during the earlier stages of the flood onset and so the social response to the flood is slow when there are fewer people in Carlisle to
disseminate the message of evacuation. This finding further reinforces the results presented
in the diffusion model (figures 8 a & b). Without a threshold number of the population being
aware of the impending flood there is less likelihood of contact with unaware agents. This
means that the response dynamics are more reliant on the innovative procedures of policy
uptake and arbitrary choice, both of which are shown to be less likely to produce a *successful*
evacuation outcome. The transition from micro to macro level response, from individual
agent interaction up to a large group response to changes in the environment, is realistically
a much more complex process than that illustrated in the HABM model. Thus, as a starting
point for testing hypotheses related to transitory-scale flood hazard response, it is a useful
tool for exploring the related and inherent complexity of the socio-environmental interface
present during a flood event (Wilensky & Rand, 2015; Wisner et al., 1994; Wong & Luo, 2005).

[Figure]

**Figure 11:** A representation of the key results shown in Figure 9 together with concepts that can be associated
with them. It is expected that these percentages will vary with model parameterisation and changes in the area
modelled.

5. **Discussion**

From further interpretation of figures 8 a & b, 9, 10 and 11 it is reasonable to infer that the
agents within the HABM, representing the local population of Carlisle, demonstrate a further
*preference* for basing their response to a flood event on interaction with their surrounding
neighbours, a social response, rather than acting directly from policy instruction. The 2005
event in Carlisle significantly overtopped existing defences, meaning that local and possibly
larger scale *management* actions would have been of little consequence to the event
dynamics and so it is here where the social response becomes influential in the risk and
resilience dynamics of the event (De Groot and Schuitema, 2012; Kinzig et al., 2013 ). With
respect to these dynamics of response, the rate of innovation (Figure 8a) impacts less of the

**Commented [TO27]:** Narrative enhancement Ref #2
Clarification of how societal actions, simulated within
the HABM, have impact on the event outcome.

**Commented [TO28]:** Technical correction Ref #1
(Figure 8a), (Figure 8b) [add the word 'figure', and not in
bold].

[revised manuscript text omitted]

**RC1                                  –                                  Response**

**The authors would like to extend their sincere thanks to the referee for their time and considered thoughts on the submission. All comments and corrections have been thoroughly considered, with our respective action and/or response to these outlined below (in red.)**

**General comments**

This paper recognises the complexity of hazard situations and responses, but also that adaptive actions overall may be simulated from individual or 'agent' behaviours through using agent-based models (ABMs). On the physical side, hydrodynamic behaviour can have an equivalent concern for the local through detailed topographic modelling and floodwater routing. The paper demonstrates how combining the two, through an innovatively developed approach coupling hydrodynamic and agent-based models (named here HABMs), allows site-specific procedures for warning provision and evacuation to be usefully designed. This is accomplished through simulating populations and exploring their alternative behaviours to see which might be of most benefit for responses to flooding events given the local geography – as in the case of Lancaster, UK, flooding described here. An interesting feature of the approach is that the human behavioural aspects are here justified by appeal to social theory, just as the (now better-established) hydrodynamic modelling is justified and rests on physical theory.

The authors appreciate the referee's acknowledgement of their attempt to innovate an approach towards the development of useful designs for warning provision and evacuation that harness the detailed elements of physical and social theory together. The authors agree that physical theory is, currently, the better-established format for justifying action with respect to warning provision and evacuation but also believe that there are influential degrees and actualities of the warning and evacuation processes which are not yet, or cannot be, accounted for within physical theory. It is here where the authors hope their attempts to illustrate the potential influence of these unaccounted factors, through the lens of urban flooding and the HABM framework, find the greatest value.

**Specific Comments**

The promotion of new quantitative approaches that combine physical understanding of hazards with possible actualities of human responses to them is surely to be welcome. Until recently there has commonly been an academic gap between the two: (1) improved modelling of physical phenomena and their dynamics on the one hand, but (2) 'top-down' imposition of (mostly hard engineering) solutions at affected sites without exploring what their populations might be doing, or could best be doing, in response. Localized decision-making is likely to improve greatly if those involved have good understanding of what best to do in the situation they confront – rather than putting schemes to the vote at some higher political level, the advantages or disadvantages of which are little understood on the ground. 'Participatory methods' have to be better than this. Coping with hazards is at heart a human cognitive activity, and so how people at different participatory levels can behave, or get informed as to how better to behave, should be beneficial.

The authors roundly agree with this assessment and hope the essence of this agreement can be felt from reading the submitted paper. De Groot and Schuitema (2012) suggest, quite robustly, that there is a distinct link between the acceptability of environmental policies, social norms and the characteristics of those policies, further to which, Kinzig (et al., 2013) suggests that the insufficient insight on the coevolution of these norms and policy instruments is what compromises the ability of decisionmakers to craft effective solutions to society's most intractable environmental problems. The authors recognise the growing annual losses attributable to the environmental problem of flooding as an extension of this lack of insight and as having a solution in the analysis, evaluation and development of participatory methods which are equally informed by both physical **and** social theory. This paper and example therein serve as a vehicle for this sentiment and the authors hope the approach outlined in the paper serve as a catalyst for the development of further hybrid narratives that are necessary for the advancement of effective participatory methods and policy.

**Technical Corrections**

The referee's direction for technical corrections throughout the submission are very much appreciated by the authors and these have been implemented within an updated version of the manuscript to be uploaded following the period of interactive discussion.

18. '... constructed using the Bass Diffusion Model'. x

118. Omit last comma in cited references. x

127. Put 'Dawson' in the bracketed reference. [also line 570] x

171. Dawson et al., 2011; Müller, . . . x

183. Semi-colons needed between EA Reports: 2006; 2012; 2016. And after Neal et al., 2009; [Also in line 752 after 2003.] No need for 'ands' within the reference brackets; see also lines 207-8; 214; 234; 267.

304. 'imitators' x

305. a priori. x

323. Chen & Zhan, 2008; x

334. One quote mark only needed. x

394. Dawson et al., 2011; also, no full stop after 1994 within the brackets. x

487. Comma after information [to be consistent with elsewhere]. x

587. Add full stop to sentence after bracket. x

627. How does the map show 'areas through which people are most likely to move' as the caption suggests? That's made more visible in figure 10. x

651. Why a semi-colon here? Perhaps: '…traditional terms that may be thought of as an acceptance'. x

654. Give cited reference, not just its number. [Also line 665]. x

659. Semi-colon needed after 2015. x

719. Last sentence of caption incomplete. x

751. Omit 'of'. x

758. Full stop after bracket, not before it. x

846/7. (Figure 8a), (Figure 8b) [add the word 'figure', and not in bold].

875. 'being based on shifted Gompertz…' x

902. 'value' rather than 'truth' perhaps.

907. Not bold. Need to check house style (especially whether 'figure' should have a capital letter). This long sentence at the start of the paragraph needs recasting, too.

923. their understanding.

940. Readers might appreciate a page number for this quotation.

Lettering sizes on Figures 2, 3 and especially the side panels of Figures 5-7 are on the small size.
* * *

* * *
**RC2 – Response**

**The authors would like to extend their sincere thanks to the referee for their time and considered thoughts on the submission. All comments and corrections have been thoroughly considered, with our respective action and/or response to these outlined below.**

This paper proposed an innovative approach to represent the complex human behaviour during flood evacuation in Carlisle by combining a hydraulic model (LISFLOOD-FP) and an Agent-Based Model (NetLogo). I have really liked the idea of using the Bass Diffusion Model to represent the agent's behaviour during flooding. The results of this study demonstrated the importance of using a holistic approach to flood management purposes. Overall, I have enjoyed reading the paper and I found the manuscript well written, clear, and results are properly described and discussed. For this reason, I do recommend a minor revision before this paper can be accepted in NHESS. However, I still have a few comments which I hope will be useful to the author to strengthen the manuscript.

The authors appreciate the referee's kind comments and recognition of our efforts to holistically frame the dynamics of flood events using a combined socio-hydrological modelling tool. In sum, we have found the referee's thoughtful comments useful in strengthening the revised manuscript, to be uploaded following this period of interactive discussion.

1) It looks to me that one important aspect of the ABM is not included in your approach, i.e. the traffic model. In fact, during the evacuation process, traffic congestions can play a crucial role before the agents select to respond to the flood all at the same time. However, it is not clear to me what are the dynamic characteristics of the agents 'movement (e.g. speed) and how are the road features included in the ABM. In fact, evacuation strategies may change based on the direction, capacity, and maximum allowed speed of the road network in Carlisle.

The authors do agree with the referee that traffic models offer an important aspect to ABMs and can have impact on the response to flooding. In the first instance, it was felt that there was already a wealth of models that had implemented traffic flows in ABMs. We wanted to focus on developing something different and whilst traffic dynamics have been implemented in the latter iterations of the HABM, this was a matter of course rather than interest and has little impact on the novelty of findings outlined in this paper. Simulations were run where the dynamics of agent movement varied between 1m/s and 3.5 m/s, to represent 'walking' up to a 'brisk pace'. The exit from the DEM is the action towards which 'warned' agents will move. Not all agents will do this, some will just move to a safe distance and then re-interact with the routine in the following time-step. This is thought to best represent the dynamics of human response that people would give to a flood like that seen in 2005 Carlisle – slow onset and propagation. The road features were implemented from open street maps and these provided the avenues upon which agents could move and interact with the environment.

2) Why did the authors couple LISFLOOD-FP with the ABM if societal actions will not influence flood propagation (at least in this study)? Of course, the proposed coupling framework can allow simulating more complex situations, e.g. placing sandbags or other tools to protect from flooding, but it will drastically increase computational costs. I assume that such costs may reduce if the raster files are uploaded within the NetLogo framework each simulation time step. Moreover, what is the computational time for 1 simulation?

This flood event was a 1 in 150-year event which significantly overtopped the existing defences meaning that local and even large-scale management actions would have been of little consequence to the event dynamics. Furthermore, we are concerned with the process of in-event, societal response to the flood propagation. Of primary interest here was the modelling of communication dynamics. Whereas placing sandbags can indeed be defined as a routine, antecedent response that influences flood propagation, we suggest that the characteristics of responsive action (the patterns of inter-agent communication and subsequent action) taken by agents to the flood and in the simulations would not be present if the flood did not happen and so is analogous to the process of innovation. Here, we are framing response by adopting the terminology used by the governmental guidelines for flood planning in the sense of human, individual and community, 'plans' and we offer some insight into how the concepts and patterns of individual and community communication and response can be represented within an ABM. The time taken to model this process, over 1 complete simulation of the flood, without any variance in the parameters and dependent on the computer system used, has ranged from 45 seconds to 3 minutes 30 seconds. We found that implementing a dynamic flood wave within NetLogo exponentially increased computation time and thus moved to importing raster files which offered relatively faster simulations, overall and at each time step, of the dynamics and interactions of interest.

3) How the working locations for all the agents are assigned? From what I could understand from the manuscript, the daily routine is randomly assigned at each simulation based on the census information of the specific commercial area in Carlisle for 2005. Is this valid also for the working locations?

The daily routine is present throughout the whole simulation for all agents to carry out. Yes, this is valid for the working locations also and is sourced from the census flow dataset.

4) When an agent receives the warning and decides to act immediately it will then exit the DEM using available network road. Is this a realistic situation? If yes, please provide a reference to support your choice.

With respect to the ABM outlined in this manuscript, we chose to develop and focus on the aspects of community, individual choice and action. This was justified through reference to the UK Government's

'personal flood plan'. To ensure that these aspects were as dynamic as possible we recognised that we needed to give the agents the choice to 'respond' to the flood propagation based on proximity to flood waters and/or on inter-agent communication but also the choice to not respond and continue with their daily routine. Being 'pre-warned' simply gives an agent the option to immediately seek an exit from the DEM as they are aware of the impending flood. In terms of this being a realistic response, the authors inferred this process of moving away from the flood waters as being realistically representative of a choice people would make based upon reference to The Environment Agency's 'Flooding: what to do before, during and after a flood' document from 2015. This will be added as a reference in the updated document.

5) Can you provide an example of the "innovative knowledge to respond to the flood upon onset" that a pre-prepared agent can use? (line 579) Maybe I have missed some details.

Upon deliberation, the authors suspect they might have explicated this in a clearer fashion for the reader. An example would here be classed as knowing how and when to leave along a particular route that leads to safety Here, we explain that the knowledge of the flood and thus the requirement to respond in a fashion which is beyond that of the daily routine is innovative in its own right, or at least is analogous to the essence of an innovation. This is different to undertaking an action which you might class as an implementation of a 'hard-engineered' innovation and is linked to the terminology of The Bass Model and Tarde's terminology for the laws of imitation. The format of human response and communication is necessarily innovative owing to the relatively infrequent unification of human and natural environments in the format of a flood event.

6) Besides for the DEFRA estimation for Carlisle at line 756-758, did you evaluate the model results with other observation data (e.g. tweets or report for some specific parts of the city)? I have found some (maybe useful) information in this webpage http://www.intrescue.info/hub/index.php/carlisle-floods-8th-january-2005/

This is a very useful source. However, it seems that the information in this source does overlap with that provided within the DEFRA reports, which were used to inform the dynamics of interaction within the HABM. We feel that the information contained in the source provided by the referee could be useful for informing and developing a sub model routine for agents who choose to remain in their properties during flood propagation. Aside from DEFRA, local and national tabloid accounts were used in cross-referencing event timelines and these were found to be useful in the absence of twitter or indeed any digital footprint of note for the event in 2005.

7) The authors stated that "The only study to date to drive an ABM with a hydrodynamic model was that of Dawson (et al., 2011)." This is not totally correct. Also, in Medina et al. (2016) an ABM and a hydraulic model were coupled to test large scale evacuation strategies in coastal cities under threat of imminent flooding due to extreme hydro- meteorological events. Moreover, other studies coupled ABM with a hydraulic model for flood risk management purposes (Abebe et al., 2019).

This statement has been revised to indicate that there are indeed other examples of ABMs driven by hydrodynamic models. In making this statement, the authors were referring to a model which they felt would be directly comparable by scale and computability, this could have been made clearer. We also feel that these references are good additions to the paper and so they have been included in the revised manuscript.

8)    Try    to    improve    the    quality    of    figures    8    and    9

Yes, this will be implemented in the revised manuscript.
* * *

* * *
**RC3 – Response**

**The authors would like to extend their sincere thanks to the referee for their time and thoughts on the submission. All comments and corrections have been thoroughly considered, with our respective action and/or response to these outlined below.**

"This paper attempts to present an integrated hydraulic-ABM model for modelling individual behaviour during flooding. Human interventions could significantly affect flood risk even during an event, especially in densely populated urban areas. This research represents an encouraging attempt to develop an approach to model human activities in the city of Carlisle during a flood event in 2005, which is an innovative and necessary step forward in flood risk assessment. But at its current form, the paper is difficult to follow, and it is not clear what the core focus and innovation is. It must be substantially revised and improved before accepting for publication. Hope the following comments will help the authors revise their                                                                                                                      paper."

Author response: The authors appreciate the referee's acknowledgement that this is indeed an encouraging attempt at developing an innovative and necessary step in the field of flood risk assessment. As outlined in the responses below, the authors have sought to address the referee's concerns and to clarify further the core focus and innovation of the paper.

**The major concerns:**

1. What is the major novelty or focus of this work? Is it the 'new' modelling framework? Or is it the application of the model to understand human activities during a flood event in the case study?

Author response: To broadly answer this series of questions, this work is an improvement on previously conducted work (e.g. Dawson et al., Lumbruso et al.) owing to: (i) the efficiency and flexibility of having two separate codes for the models, thus increasing the likelihood of the coupled model framework representing a more sophisticated set-up (inertial wave, 1D/2D structure for channel representation etc.) and (ii) having a hydraulic model that has been more thoroughly validated than models previously written into NetLogo. With respect to Lumbruso et al.'s paper, the Life Safety Model did not test the evacuation characteristics for 'type' of response. The focus of our work is to address these two shortcomings by offering a modelling approach which couples physical and social models where agents have a probabilistic daily routine and a choice of responses on an individual basis. This enables the exploration of different hypotheses for social reactions and responses to the detailed, accurate and dynamic physical outputs generated by LISFLOOD-FP by adding the related elements of policy and systematic change.

Specifically, we use the Bass Model of diffusion (l. 220-224) to explore hypotheses relating to flood warning and evacuation which yields interesting new insights into these processes that would be difficult to achieve in any other way.  It follows from this that the framework is indeed new and by applying it to the case study for Carlisle's 2005 event we are able to illustrate human activities and understand their behaviours, structured with a logical and believable social model and driven by a firmly validated physical model.  We therefore believe the work has a clear focus and is novel in endeavour, as was noted by the two other referees.

"This paper presents a new flood risk behaviour model developed using a coupled Hydrodynamic Agent-Based Model (HABM)", which suggests the modelling framework is the key novelty in this work. But the presented HABM takes offline modelling outputs (flood depth) from LISFLOOD-FP to drive the agent-based model developed in the NetLogo framework. This is a 'step backwards' from the modelling approach as reported by Dawson et al. (2011), in which "a hydrodynamic model simulates the floodwave was also developed within the ABM platform and interacts directly with the agents and the built environment".

Author response: Concerning the (excellent) work by Dawson et al., we argued in the paper that "this study initially coded the hydrodynamic model directly within the ABM meaning advantage could not be taken of recent developments in efficient numerical methods for solving the shallow water equations ... and high-performance computing…" The fundamental thought to this is that the approach taken by Dawson et al. was a great way to start to link ABMs and hydrodynamic models, but we found that it has some technical limitations because only a very simple hydrodynamic model can be coded within the ABM framework. The referee has perhaps not appreciated the limitations imposed by writing the hydrodynamic code within the ABM, so these are further outlined below:

Because they were working within the NetLogo ABM framework, Dawson et al were only able to code a very simple inundation model for 2D only domains. This was based on solving a version of the diffusion wave equations following Bates and De Roo (2000) which was (just about) adequate for the small coastal flood that Dawson et al simulated. The coding environment in an ABM framework can never be as flexible and computationally efficient as writing software in a compiler language, as we found when we tried to do exactly this at the start of our project. We initially coded our hydraulic model within NetLogo exactly as Dawson et al had done, but for the high-resolution whole city-scale test case used here the simulation took days of computer time. This is because solving dynamical equations on fine grids with numerical methods without a compiler language is extremely slow.

In addition, the lack of coding flexibility within ABM frameworks means that one cannot create more sophisticated model structures, such as hybrid 1D/2D hydrodynamic models, that are required to simulate fluvial flooding in urban areas. The only reason for having the hydraulic model coded within the ABM is if the behaviour of the agents changes the development of the inundation. This is not the case for the Carlisle flood, and neither was it the case for the coastal flood simulated by Dawson et al. In these circumstances there is no advantage to the 'tightly-coupled' approach and it also means that one is not able to take advantage of the latest development in hydraulic modelling. For example, we showed during a series of papers during the 2000s (Hunter et al., 2005; Hunter et al., 2008; Bates et al., 2010) that the simple diffusion wave approach used by Dawson et al suffers from a series of technical flaws meaning that to correctly simulate wave dynamics it can only be used with relatively coarse numerical grids. This is problematic for simulating floods in urban areas where it is now commonly accepted that one needs a model grid capable of resolving flow around buildings. By writing their hydrodynamic code within the ABM framework Dawson et al's approach could not be used to simulate a whole city scale inundation event at high resolution as we do here. By keeping the ABM framework and hydrodynamic model separate we effectively solve this problem.

As a result, writing a hydraulic model within the ABM framework has no advantages for many (perhaps most) flooding applications and leads to quite a few constraints. Our approach is a step forward because it can use a more sophisticated hydrodynamic model that takes advantage of nearly 20 years of numerical developments since the Bates and De Roo (2000) formulation implemented by Dawson et al. Having an offline model is much more flexible and it can therefore be applied to a breadth of different situations to test different hypothesis, not just simple 2D coastal problems at relatively coarse resolution.

"The approach of using offline flood modelling outputs to drive an agent-based model has also been reported in the literature, e.g. Lumbroso et al. (2011) developed a life safety model to estimate risk to people imposed by dam breaks or flash floods. In their work, their Life Safety Model could use outputs from any available two-dimensional hydrodynamic models that solves the shallow water equations (e.g. Telemac-2D, TuFlow) or the simplified forms (e.g. LISFLOOD-FP)."

Author response: The authors acknowledge the referee's assertion that Lumbroso et al's work on the Life Safety Model offers a similar level of physical modelling flexibility to that of the HABM and thank the referee for drawing our attention to this. As far as the authors are aware this is one of few (Dawson's being the other) comparable modelling studies to the HABM described in our paper and we have included an acknowledgement of this in the revised manuscript.

There are clear differences in the two overall approaches. Lumbroso's model considers the notion of 'fate' based upon 'warning' and 'action', claiming to consider the notion of direct or indirect warning i.e. agent communication, in the process of warning or action. It does not substantiate the process of message adoption or suggest how this might better align with current policy direction on an individual level. There is no clarity on whether the agents carry out a routine of any kind, with the choices being given to them largely relying on linear and limited choice direction. We imagine the natural counter to this might well be to draw attention to Dawson having included a routine, with comparable physical modelling flexibility and here the HABM differs again by offering agents the choice of adopting an 'emergency routine' in addition to the standardised daily one. This means that the HABM emphasises the role of choice and models it in a more representative manner than in previous work.

Thus, in sum, with contemporary policy moving towards a more integrated approach this framework utilises the methods and conclusions of these two previous pieces of research and builds on them, adding enhanced theory and the necessarily enhanced methods, to provide an integrated approach to test new hypotheses; contributing to the overall sense of novelty.

"If the focus of the paper lies in the application of the model to understand flood-driven human dynamics in the case study. There is no strong evidence showing the model settings reflect reality and so the results and the conclusions may be misleading."

Author response: The authors would like to direct the referee's attention to the cited paper by Neal (et al., 2009) regarding this point. The primary reason why this case was chosen is because of the quality of the computer model used in the simulations. This is also covered sufficiently in figures 9 and 10 of the paper, specifically in (l.745-779) it is stated that over the simulations conducted, the number of potential casualties was aligned with that which was actual during the event in 2005. However, upon review this could be made clearer and so we wish to assure the referee that this has been done for the final submission.

2. Following the above comments, it is difficult to be convinced that the model settings can represent actual human dynamics during a flood event in Carlisle since:

Author response: The purpose of this paper is to test hypothesis (l. 144-148) and in respect of this, the human dynamics that the ABM simulates are sufficiently 'real' to produce results which are in line with those observed during the event modelled. It is also the case that all models are a simplification, but here, we believe the HABM represents suitable complexity for the scientific purposes to which it is being put.

i) the behaviour rules for individual are over-idealised and there is no evidence to back the choice of behaviour rules;

Author response: The behaviour rules are directly sourced from Dawson et al. and, upon reflection, are no more idealised than the responses seen in Lumbroso et al. As an example, in Lumbroso's paper there is no justification given for the scalar magnitude of diffusion of choice (i.e. the effect of choices made by agents, on other agents) and, where alluded to, it is not founded in the kind of arguments we outline in sections 2.3, 2.4, 3 and 4 (l. 721-729) of this paper. Again, Dawson (et. al)'s model, which is another paradigm of physical modelling, makes no substantive reference to social system representation beyond that which is basically necessary for coupled analysis. Further, with respect to agency routine, the authors would argue that Lumbroso's 'PARU' approach is more idealised in comparison to that of the framework in this paper.  This particularly being so when there is little information given with respect to how these (PARU) units form and no detail given with respect to the process of choice in the formation of these 'evacuative' units. In our case the interaction rules within the HABM are based on laws of sociological diffusion (Larsen et al., 2005 – source added to revised submission), which take the agents through the five steps of Gabriel Tarde's law of imitation and invention. These are terms which are much better aligned with the reality of what behaviours individuals are likely to exhibit in social settings than anything the authors have reviewed during the process of the model development, or since.

The authors did refer to Bernadini (Bernardini et al., 2017 – source added to revised submission) during the initial stages of developing the behavioural rules alongside the framework provided by Dawson et al. as well as the Nomis and Flow data sets which were further used by the authors of this submission as a cross-reference. Combined, these sources gave rise to the general routine presented in the paper. It is hoped that with this clarification and with the additional source materials added, the referee will see that the choice of behaviour rules and routine are grounded in both legitimate evidence and theory.

ii) the communication rules between agents are also over- simplified, e.g. how are text, social media and other forms of wireless communications taken into account, which may significantly affect the simulation results;

Author response: Whilst being 'en vogue' currently, this is not the chosen focus of the paper and also, during 2005 this was much less of a factor for consideration than it is today as many networks for these forms of communication were still being developed. The 2015 Carlisle event would provide an interesting contrast to 2005 as it would be a model within which such formats for communication would presumably provide mensurable impact and thus would merit inclusion in upcoming models and study. We again stress that in the paper we are trying to test several hypotheses concerning flood warning and response, and not produce an exact facsimile of the real world.  All models simplify to some extent and we would argue that this is reasonable evidence that we have included enough complexity in our model to undertake the science objectives of the paper.

iii) traffic systems and key organisations are not represented in the model which will inevitably have significant influence on the results and conclusions;

Author response: Yes, potentially they may have influence for conclusions linked to evacuative action but as is stated in this paper, the significance may be allocated at the outset of process i.e. how warning is communicated rather than how action is taken. We again note that the physical and human dynamics included in the model were chosen based on theory with a strong lineage of scholarship from other disciplines in order produce a new platform for experimentation and interpretation and practice. In this respect our view is that the HABM delivers with effect.

iv) the model results were not validated at all. Therefore, the results and the conclusions from the simulation may not be valid and may be misleading.

Author response: Were the aim of the work to make predictions and/or forecasts then yes, further validation would not only be imperative but of great value in addition to the aims and scope of this paper. However, to further reassure the referee, the authors are confident that the hydraulics modelled are well validated for the Carlisle 2005 case study, as is supported by the large body of cited works in section 2.1 of the submission and that the human dynamical routine is eminently sensible and realistic (sufficiently so to answer the questions posed in the paper).  Additionally, and as the referee will be aware, ABMs are historically difficult to validate (Ormerod and Rosewell, 2006 – source added to revised submission) and whilst techniques have been introduced to improve this, the authors feel that the model offers a sufficient balance between " clear explanation and description of the phenomena" and the "simplest possible realistic agent-rules of behaviour" for the model to be considered a valid base for comparison to other models (such as those suggested by the referee i.e. Lumbroso et al. & Dawson et al.)

The authors would also argue that the level of cognition afforded to the agents operating within the model is not so high as to require significant justification beyond that provided as the process represented is of sufficient alignment to produce useful results for an intended purpose, namely to test hypothesis which would be difficult to evaluate in any other way.

Minor issues:

Author response: These issues are a precis of those outlined above and so have largely been addressed above.

1. Why the authors use the 2005 flood event but not look at the more recent 2015 event? More information would be available from different sources for the more recent event to inform and validate human activities.

Author response: As stated in the paper, the 2005 event is one which has provided a large amount of data from LISFLOOD and resulted in a large body of published information on the related phenomena. On this basis, it was felt that it provided a suitable, initial, case study for the application of the new framework – as stated in the submission. Furthermore, as stated in 2 (ii), the 2015 event will provide excellent scope for an updated model which will include the new formats for communication.

2. The paper is difficult to follow, and the authors should more explicitly explain the modelling framework, how the agents are interact(ing) and communicat(ing), and how the behaviour rules are set and why, etc.

Author response: At 32 pages, the authors feel that they have invested enough time and care to ensure the framework of the model, the formats of interaction and communication and the setting of behaviour rules are all explained in enough detail. Where necessary, we have provided further source material for the reader's reference to consolidate this detail.

3. Since the human activities do not have any impact on the flood dynamics and the agent-based model is only driven by offline flood model outputs, it is NOT a 'coupled' model.

Author response: As has been emphasised in the author's responses to all preceding assertions made by the referee, the key and novel difference of this submission is the development of a framework that offers scope to include steps seen in directly coupled models (of the same nature) as well as scope for including indirectly coupled procedures for modelling interactions from beyond the scope of those models (of different natures). The motivation here being a desire to move towards more inclusive narratives that align with the dynamic notions of vulnerability and transcend the infinite regress of 'risk-based' modelling simulacra, which seemingly feed into the 'Tower of Babel' problem and do not seem to be addressing the issues of growing disparity in modelled and realised loss; nor incorporating the growing movement in policy to incorporate fundamental elements of social science (l. 79-84 in the submission). Ultimately, were the models not coupled, no results would have been produced to represent the different aspects modelled i.e. the flood layers called into the model would not drive any response in the agent population. Therefore, the authors believe this to be associated with semiotic misunderstanding and so will move to clarify this in the final submission.

4. The title, 'an agent-based model for flood risk warning', is a bit confusing. Based on its current capacity, the model cannot be used for 'flood risk warning'.

Author response: Without a suggestion for an alternative we are unable to consider what might be a better alternative. In the most basic format, based on the physical representation of the flood and the subsequent modelled response of the population in the model, this is an agent-based model for flood risk warning.
* * *
**Referees references:**

**Dawson, R., Peppe, R. & Wang, M., 2011, An agent-based model for risk-based incident management of Natural Hazards. Nat. Haz., 59(1): 167-189.**

**Lumbroso, D.M., Sakamoto, D., Johnstone, W.M., Tagg, A.F. and Lence, B.J., 2011. Devel-opment of a life safety model to estimate the risk posed to people by dam failures and floods. Dams and Reservoirs, 21(1): 31-43.**
* * *
Author's                                                                                                       references:

Neal, J. C., Bates, P. D., Fewtrell, T. J., Hunter, N. M., Wilson, M. D. & Horrit, M.S., Distrubuted whole city water level measurements from the Carlisle 2005 urban flood event and comparison with hydraulic model simulations. Journal of Hydrology, 42-55, 2009.

Larsen, G.D., Horses for courses: relating innovation diffusion concepts to the stages of the diffusion process. Journal of Construction Management and Economics, 23 (8), 787-792, 2005.

Bernardini, G., Camilli, S., Quagliarini, E. & D'Orazio, M., Flooding risk in existing urban environment: from human behavioural patterns to a microscopic simulation model. Proceedings from the 9[th] International Conference on Sustainability in Energy and Buildings, SEB-17, Chania, Crete, Greece, 5-7 July 2017, Energy Procedia, 134, 131-140, 2017.

Ormerod, P. & Rosewell, B., Validation and Verification of Agent-Based Models in the Social Sciences. In: Squazzoni, F., Epistemological Aspects of Computer Simulation in the Social Sciences. EPOS 2006. Lecture Notes in Computer Science, Springer Berlin, 5466, 130-140, 2009.

---

## Referee Report (RR1)

Unfortunately, most (all) of my comments provided in the discussed version were not addressed. As it stands, this work does not present any major novelty, does not provide any major values to the existing literature in this topic and furthermore, the results and conclusions may be misleading. I cannot recommend accepting the paper at its current form. The major comments are summarized once again here:

- 1. Lack of major novelty: "This paper presents a new flood risk behaviour model developed using a coupled Hydrodynamic Agent-Based Model (HABM)". What is new here? The flood inundation model has been used for over a decade; creating such an ABM model on NetLogo is a straightforward task and no major advance is presented, compared with e.g. the one presented by Dawson et al. (2011) which was also reported almost one decade ago. "Instead of directly embedding the hydrodynamic model within the ABM, a more pragmatic solution is to indirectly couple a separate, and highly optimized, hydrodynamic model with an existing ABM framework." This is a bizarre statement/argument which does not explain why we don't need to couple the models and can't take advantages of the inundation model by 'properly' coupling them together. HABM "uses water depth output files from the LISFLOOD-FP at each model time-step within a simulated version of the affected area". Whilst it is presented as 'a coupled' model, the two modeling components are not even integrated together and the HABM simply uses the results from LISFLOOD-FP to inform the agent behaviours. There is no interaction between the two modelling components. The authors should stop exaggerating their work or model and use correct terminology. So, the model(s) as presented are not new and actually the whole paper lacks major novelty.
- 2. The model adopts oversimplified behaviour rules to drive the interactions between agents and does not consider major 'actors' that play key roles in flood evacuation and risk propagation processes and so will not be able to provide any meaningful results to address the "two currently unresolved questions relating to flood evacuation warnings" as claimed. For example, the transport systems and all of the relevant government agencies or organisations are not included. Again, the authors provide an unconvincing augment for this, "discrete transport model was not included in this model for these initial findings as it was felt that there has already been recent and significant advances in this area of interest". But transport systems are a key 'actor' in any of the flood evacuation/flood impact model related to population and must be taken into account to ensure the results are presentative! Modelling individual behaviours have also been made 'significant advances' recently. Why the authors bother to present this work then?
- 3. The model does not consider sufficient social processes during a flood event to ensure the modelling results to be representative and meaningful. Also, the model has not been validated by any means. The results being presented and the following conclusions are likely to be misleading, and certainly do not help address the 'two currently unresolved questions relating to flood evacuation warnings" as claimed.

---

## Author Response (AR2)

**SCHOOL OF GEOGRAPHICAL SCIENCES**
University of Bristol
University Road
Bristol
BS8 1SS
UK
Tel: +44 (0)117 928 9954
Fax: +44 (0)117 928 7878
E-mail: geog-office@bristol.ac.uk
www.ggy.bris.ac.uk

*From*:        Tom O'Shea
*E-mail:*      t.oshea@bristol.ac.uk

Dr. Carmine Galasso
Editor
Natural Hazards and Earth System Science

**Re. Author response for O'Shea et al.**

Dear Dr. Galasso,

Many thanks for your e-mail of 1st May 2020 enclosing referee comments on the above paper, for which we are very grateful.  We are pleased that referees #1 and #2 are now happy to accept the manuscript and we have undertaken further revision of the paper to address the outstanding concerns of referee #3.  In this we have been guided by the helpful advice of referee #2.

Specifically, we have included in the manuscript a description of the differences between our study and previous ones on a similar topic.  We have also included text from our previous reply to referee #3 in the manuscript as referee #2 suggested.  We are really pleased that referee #2 appreciated these comments.  We hope with these changes the paper will now be acceptable for publication.

Point by point responses are given below (referee comments in black, our response in red).  All line numbers given in the revision letter refer to the new 'tracked changes' version of the manuscript.

Yours sincerely,

Tom O'Shea

**Editor Decision: Reconsider after major revisions (further review by editor and referees) (01 May 2020) by Carmine Galasso**
**Comments to the Author:**

Dear authors, many thanks for having submitted your revised manuscript and for having addressed the reviewers' comments.

As you can see below, Reviewers #3 wasn't satisfied with your response/revised manuscript. However, both Reviewers #1 and Reviewers #2 have recommended acceptance of the manuscript.

One of the reviewers suggests some revisions in light of the comments raised by Reviewer #3, to better highlight the novelty of your study.
Thanks, we have followed this advice and also implemented other changes to better address the concerns of Reviewer #3.

Also, both I and an Executive Editor have noticed some poor presentation of illustrations (e.g., figures 5-6-7 without the appropriate legend of colours, missed scales; fig. 8 with a strange format according to international standard, etc). Please improve the quality of your figures.
Thanks for picking this up. These changes have now been made. With specific reference to the editor and referee's requests for better quality figures, there are instances where this was not possible without altering the orientation of the pages and moving these figures to an appendices, to accommodate the dimensions required for improved figure quality. Figure 1 was changed to a higher quality QGIS map, figures 5-7 have been enhanced significantly. Figures 8 a & b have been rectified according to the specifics of the requests and figure 9 has been enhanced where possible so that the finer details are not clearer.

Many thanks again for your contribution. I look forward to receiving a revised version of your manuscript.
Thanks to you and the Executive Editor for handling the review process!

Best wishes
Carmine Galasso

**Reviewer #1**
No changes required.

**Reviewer #2**
I would like to thank the authors for their excellent work in answering the reviewers' comments.
Thanks, this is very kind of you.

However, I would suggest including in the manuscript a description of the differences between their study and the previous ones on a similar topic (e.g. Dawson et al., 2007 and Lumbroso et al., 2011). I really appreciated the way the authors replied to reviewer #3 and I would recommend including those replies in the manuscript so that other readers with the same doubts could better understand the novelties of your study.
Thanks, this is a very good suggestion which we have now implemented.

Besides that, I do not have any further comment to make. I wish the authors all the best for their future research.

Thanks for all your constructive feedback which has significantly improved the manuscript.

**Reviewer #3**

An agent-based model for flood risk warning – O'Shea et al.

Unfortunately, most (all) of my comments provided in the discussed version were not addressed. As it stands, this work does not present any major novelty, does not provide any major values to the existing literature in this topic and furthermore, the results and conclusions may be misleading. I cannot recommend accepting the paper at its current form. The major comments are summarized once again here:

Thanks for this comment. We have clearly failed so far to convince you that our work is novel and that the results have value. We have therefore undertaken further changes to manuscript to try to address these concerns as detailed below.

We believe we did address all of the comments in our previous response, but these did not all result in changes to the text. This is because we respectfully disagreed (and continue to disagree) with a number of the reviewer's assertions. In our previous revision letter, we presented what we believe are sound arguments that justified our interpretation of the work. It follows that if one accepts this position, as referees #1 and #2 have clearly done, then changes to the manuscript may not in fact be necessary.

1. Lack of major novelty: "This paper presents a new flood risk behaviour model developed using a coupled Hydrodynamic Agent-Based Model (HABM)". What is new here?

The novelty is the use of the framework to test hypotheses about flood warning communication derived from the Bass Model of diffusion. This set of hypotheses has not previously been studied in this context and the HABM provides the mechanism for doing this. We think the referee has misunderstood the purpose of the paper and believes this to be the development of the modelling framework. The HABM is, in fact, only a means to an end, and the originality in the paper lies in the hypothesis testing. It would therefore not be an issue if there was nothing at all novel about the framework. The key advance is the use to which it is put.

We have changed the title and abstract of the paper to clarify this. We have also changed text on lines 160-161 to further emphasise the point.

Having said this, we do think there are several reasons why the HABM framework we use has some elements of novelty, but these are not central to justifying why this study is a new contribution to science and should be published.

The flood inundation model has been used for over a decade; creating such an ABM model on NetLogo is a straightforward task and no major advance is presented, compared with e.g. the one presented by Dawson et al. (2011) which was also reported almost one decade ago.

We are afraid that this is simply not the case. The hydrodynamic model is significantly more complex (both in terms of physics and resolution) than that used in Dawson et al and it is rigorously validated for the specific test case we study. Each of these elements is very clearly an advance on previous work. One can debate how big a step either is, but as we note above the development of the framework is not the main focus of the paper, which is to use a suitable numerical experiment to test hypotheses about flood warning communication.

To make this clear we have expanded the rationale for our approach on lines 126-164 to include the arguments from our previous response as suggested by reviewer #2.

"Instead of directly embedding the hydrodynamic model within the ABM, a more pragmatic solution is to indirectly couple a separate, and highly optimized, hydrodynamic model with an existing ABM framework." This is a bizarre statement/argument which does not explain why we don't need to couple the models and can't take advantages of the inundation model by 'properly' coupling them together.

Again, we respectfully disagree. There is nothing bizarre about this approach. As we previously argued, the only time the 'tightly coupled' model set up that the referee regards as the 'proper' approach is needed is if the agent behaviours can change the propagation of the flood. When the ABM represents the general population, this is very unlikely to be the case, and certainly did not occur for the Carlisle 2005 event. We agree that if the ABM represented specific risk management authorities who could, for example, choose to operate relevant flood control infrastructure then the approach advocated by the referee would be needed. However, because we here test the impact of different types of warning communication on the behaviour of the general population a one-way interaction is sufficient. In this case the flood simulation is simply a boundary condition for the ABM.

The approach advocated by the referee thus has no advantage in our case (and likely in many others) and a number of serious disadvantages such as reduced flexibility. However, as we note above, and have stated in our previous response letter, the framework is not the key focus of the paper.

Lines 126-164 now make this point in a way that is now hopefully absolutely clear using the arguments from our previous response as suggested by reviewer #2.

HABM "uses water depth output files from the LISFLOOD-FP at each model time-step within a simulated version of the affected area". Whilst it is presented as 'a coupled' model, the two modeling components are not even integrated together and the HABM simply uses the results from LISFLOOD-FP to inform the agent behaviours. There is no interaction between the two modelling components. The authors should stop exaggerating their work or model and use correct terminology. So, the model(s) as presented are not new and actually the whole paper lacks major novelty.

No, there is very clearly a one-way interaction: the flood propagation can affect the behaviour of the agents but not vice-versa. This is very clear in the text, and we do not present exaggerated claims. Moreover, this one-way interaction is exactly what happened during the Carlisle event when considering the behaviour of the general population.

We now directly address the issue of two-way versus one-way interaction (aka 'tight' or 'loose' coupling) on lines 147-150. We have also added a caveat to the conclusions on lines 1074-5 to reiterate this point.

The key novelty of the work is the hypothesis testing and not, as the referee persists in believing, the model framework itself. We believe that elements of the framework are indeed novel, but this is not the key point of the paper.

2. The model adopts oversimplified behaviour rules to drive the interactions between agents and does not consider major 'actors' that play key roles in flood evacuation and risk propagation processes and so will not be able to provide any meaningful results to address the "two currently unresolved questions relating to flood evacuation warnings" as claimed. For example, the transport systems and all of the relevant government agencies or organisations are not included. Again, the authors provide an unconvincing augment for this,

"discrete transport model was not included in this model for these initial findings as it was felt that there has already been recent and significant advances in this area of interest". But transport systems are a key 'actor' in any of the flood evacuation/flood impact model related to population and must be taken into account to ensure the results are presentative! Modelling individual behaviours have also been made 'significant advances' recently. Why the authors bother to present this work then?

With respect, the behaviour rules we present are not oversimplified and follow logically from previous work in this field (Dawson et al., 2011; Bennet and Tang, 2017). All models simplify, but the approach taken here provides sufficient complexity that we can isolate and test specific aspects of flood warning communication and draw conclusions about these. As we note in the conclusions, we are not developing a predictive tool but are instead using a model framework to gain insights that could not be obtained in any other way. The results are scientifically interesting because the Bass Diffusion model has not previously been used in this context. Our review of past work has yielded only two previous papers in this field (Dawson et al and Lumbroso et al) and it seems unreasonable to believe that there is now nothing new that can be said on this subject. We make no apologies for being clear about where future work can improve on what we present, but this does not mean that our conclusions are not a novel contribution. We think the section on behaviour rules (lines 540-586 and Figure 4) is already sufficiently clear and have therefore not implemented further changes at this point.

3. The model does not consider sufficient social processes during a flood event to ensure the modelling results to be representative and meaningful. Also, the model has not been validated by any means. The results being presented and the following conclusions are likely to be misleading, and certainly do not help address the 'two currently unresolved questions relating to flood evacuation warnings" as claimed.

As we have previously noted, the hydrodynamic model component has been extensively validated for the Carlisle 2005 test case. Please see:

Neal, J. C., Bates, P. D., Fewtrell, T. J., Hunter, N. M., Wilson, M. D. & Horrit, M.S., Distrubuted whole city water level measurements from the Carlisle 2005 urban flood event and comparison with hydraulic model simulations. Journal of Hydrology, 42-55, 2009.

Whilst idealized, the agent's daily routine portrayed in Figure 4 of the paper is clearly broadly realistic and the referee is not specific about what other processes need to be included. Point 3 is simply an assertion that is not supported by an argument. No change here is required.

**Testing the impact of direct and indirect flood warnings on population behaviour using an agent-based model.**

Commented [TO1]: New title.

[revised manuscript text omitted]
. The coding environment in an ABM framework can never be as computationally efficient as writing software in a compiler language and solving dynamical equations on fine grids with numerical methods can therefore be extremely slow.  In addition, the lack of coding flexibility within ABM frameworks means that one cannot create more sophisticated model structures, such as hybrid 1D/2D hydrodynamic models, that are required to simulate fluvial flooding in urban areas.  The only reason for having the hydraulic model coded within the ABM is if the behaviour of the agents changes the development of the inundation.  In this situation it would be necessary to have the agent behaviour and flood dynamics co-evolve during the simulation and this two-way interaction can only be achieved by having the hydrodynamic model and ABM tightly coupled in the same code.  However, this is typically not the case when the agents in the model represent the general-public rather than specific flood management actors, and for this situation a one-way coupling is sufficient. Writing a hydraulic model within the ABM framework for these cases as no advantages therefore for many (perhaps most) flood types and leads to quite a few constraints.

Commented [TO2]: Further explication added to address referee 3's concerns.

As a result, in the 'tightly coupled' approach of Dawson et al (2011) the computational costs were high, and this limited the domain size and resolution of the modelling that could be undertaken.  Instead of directly embedding the hydrodynamic model within the ABM, a more pragmatic solution when considering agents whose behaviour cannot affect the flood evolution is to indirectly couple a separate, and highly optimized, hydrodynamic model with an existing ABM framework.  This allows each code to be properly optimized for the task it performs and enables each to be more easily updated as new methods become available. This is the approach taken here, where we develop such a coupled hydrodynamic model/Agent-Based model framework (hereafter termed a Hydrodynamic Agent-Based Model, or HABM) and use this to address two currently unresolved questions relating to flood evacuation warnings.  These two specific questions are:

[revised manuscript text omitted]

**Commented [TO4]:** Figures 5-7 have been moved to the appendices to accommodate the request for better quality images.

to respond. The coefficient (**q**) is typically represented by a much smaller value than 30% in traditional applications of the model (Mahajan, Muller & Bass, 1990). However, owing to the elevated risk involved in adopting, or not adopting, the product of evacuative knowledge during a hazard scenario, the traditionally small value of (**q**) has been scaled up significantly. This is to represent a one-third likelihood (~ 30%) of those who encounter the innovator (**p**) agents, receiving the flood warning by communication and adopting directly from them. Whilst this is a manipulation of the Bass Model function, it remains consistent with the Bass Model theory, stipulating that human adoption of a process or product is more likely to happen based upon internal systematic influence, or *imitation*, rather than through external influence on the social system, or by *innovation.* Wherein the available choices may be reduced to 'yes', 'no' and 'maybe', probabilistically represented as roughly one-third each for a given scenario (Dechter & Pearl, 1986; Hart et al., 1968; Hornor, 1998; Mahajan et al., 1990; Massiani & Gohs, 2015; Sultan, Farley & Lehmann, 1996).

The fundamental difference between (**p**) and (**q**) is generated from this external-internal distinction. Aligning this further with the sociological notions of Tarde, (**p**) is a representation of an external factor that requires a change in operation of the internal system dynamics (**q**) over time, thought of as an attunement, harmonisation or, in more traditional terms that may be thought of as an *acceptance* (Tarde, 1903). This means that for an innovative process (**p**) to become a naturalised component of the internal system dynamics (**q**), a significant amount of time may be required for innovation to lead to imitation when there is a *risk* involved ( Wheater, 2006). In this application, the Bass model gives an indication of this duration based on the relative probabilistic magnitudes of (**p**) and (**q**) for a population of 108,000 agents. The overall significance of this application is that it allows conclusions to be made as to how influential external policy protocols are for the population in relation to their internal 'sense' during flood event response (Massiani & Gohs, 2015; Sultan et al., 2003).

The curves illustrated in figures 8a and b are the separate curves for the process of adoption based upon the optimised Bass Model values for the coefficient of innovation (**p**) at 50% and coefficient of imitation (**q**) at approximately 30% over 200,000 simulations for the Carlisle model. The three separate lines are illustrations of the three different iterations of the model's standard differential equation as functions of continuous and discrete time (Bass, 1969). Correspondence between the curves represents an *agreement* between the model's functions and the data being plotted. Broadly, the curves show that the innovation of external directive, seen in figure 8a (**p**), is more effective at promoting an immediate process of evacuation as a lower number of the simulated population changing state over time would suggest that a large proportion of the original innovators choose to act in the early onset of the flood and evacuate the area without hesitation. The negative aspect of this function is that there will be less agents available to communicate the innovative process and influence the less prepared agents and so this process of innovation will take longer to diffuse throughout the agent population leading to less agents taking appropriate action over a longer duration of flooding, exposing themselves to potential danger.

The curve for figure 8b, (**q**), is the internal function for evacuative measures, which is reliant on agent-agent interaction and suggests that the internal dynamics for the adoption of evacuative measures, that is to say the adoption of the same actions as the agency directive but not directly from the external directive (e-mail, text alerts etc.), according to communication and contact between agents, within the total flood affected population of Carlisle, is more influential over a shorter duration than the operation of (**p**). The variance between the three lines would suggest that there is some disagreement between the baseline functions of the Bass Model differential equation and those for discrete and continuous time functions for (**q**) and it is believed that this is likely related to the unusually high value attributed to the 30% likelihood of agents *agreeing* to imitate the innovative agents and become imitators, as well as the general stochasticity related to the reliance on 'proximal contact' for communication between agents, which is likely but not guaranteed in any situation; particularly in one as potentially frenetic as that involving a flood.

[Figure]

[Figure]

**Figures 8 a & b:** Example Bass diffusion curves for p or innovation (top), and q, or imitation (bottom), at Carlisle during the 2005 flood. Shown is the type of knowledge and subsequent action taken based upon choices made by agents acting within the HABM.

Commented [TO5]: Figures 8 a & b have been edited according to the request for revisions.

[revised manuscript text omitted]

**Appendices**

[Figure]

**Figure 1:** The total area of interest at Carlisle. An approximate area of 10km$^2$ was simulated in the HABM modelling runs. (QGIS, 2020.)

[Figure]

**Figure 5:** An overview of the preliminary HABM. Shown here as an example are agents engaging in the daily routine (green) prior to the initiation of the LISFLOOD-FP flood inundation. These figures represent only a small proportion (<1000 agents) of the full agent populations (~ 108,000 agents) simulated in the final model run.

[Figure]

**Figure 6**: Agents marked in red have become aware of the incoming flood and are taking evacuative action. Changes in agent colour on the GUI (Graphic User Interface) indicate that members of the sample population are transitioning to a 'potential casualty' as the flood encroaches their vicinity but also that the likelihood of casualty occurring will diminish over time as the message of 'preparedness' diffuses through the population.

[Figure]

**Figure 7:** Further to preparedness and potential casualty, an indication of areas in which  agents are likely to stay, areas from which  they are most likely to move as well as the areas through which  they are most likely to pass may be observed within the HABM GUI. Explicated  further in **figure 10**.

[Figure]

**Figure 9:** Box plot illustrating the range of values, sampled from 1000 agents (the most computationally stable sample size for batch runs on the available architecture) within the full agent population (108, 000), for the total number of potential casualties vs. % of population pre-warned for Carlisle over 200,000 simulations.